# Structural insights into the mechanism and dynamics of proteorhodopsin biogenesis and retinal scavenging

Stephan Hirschi [1,2] ✉, Thomas Lemmin [1] ✉, Nooraldeen Ayoub [1], David Kalbermatter[1], Daniele Pellegata[1], Zöhre Ucurum[1], Jürg Gertsch[1] & Dimitrios Fotiadis [1] ✉

Microbial ion-pumping rhodopsins (MRs) are extensively studied retinal-binding membrane proteins. However, their biogenesis, including oligomerisation and retinal incorporation, remains poorly understood. The bacterial green-light absorbing proton pump proteorhodopsin (GPR) has emerged as a model protein for MRs and is used here to address these open questions using cryo-electron microscopy (cryo-EM) and molecular dynamics (MD) simulations. Specifically, conflicting studies regarding GPR stoichiometry reported pentamer and hexamer mixtures without providing possible assembly mechanisms. We report the pentameric and hexameric cryo-EM structures of a GPR mutant, uncovering the role of the unprocessed N-terminal signal peptide in the assembly of hexameric GPR. Furthermore, certain proteorhodopsin-expressing bacteria lack retinal biosynthesis pathways, suggesting that they scavenge the cofactor from their environment. We shed light on this hypothesis by solving the cryo-EM structure of retinal-free proteoopsin, which together with mass spectrometry and MD simulations suggests that decanoate serves as a temporary placeholder for retinal in the chromophore binding pocket. Further MD simulations elucidate possible pathways for the exchange of decanoate and retinal, offering a mechanism for retinal scavenging. Collectively, our findings provide insights into the biogenesis of MRs, including their oligomeric assembly, variations in protomer stoichiometry and retinal incorporation through a potential cofactor scavenging mechanism.

Microbial rhodopsins are a large family of integral membrane proteins comprising light-activated ion pumps, channels and photoreceptors[1,2]. In addition to chlorophyll-based light-harvesting systems, retinal-dependant microbial ion-pumping rhodopsins (MRs) represent the only other known type of phototrophy to have evolved and are about three times more abundant than photochemical reaction centres[3,4]. Rhodopsins are found in all domains of life and share a common structure, which comprises a bundle of seven transmembrane α-

helices (TMH) and a covalently linked retinal cofactor[2,5–7]. The green-light absorbing proteorhodopsin (GPR) was originally discovered in marine γ-proteobacteria as the first bacterial rhodopsin and as an example of a new type of bacterial phototrophy[5]. Since then, GPR has been well characterised in terms of function and structure, and has become a model protein for MRs[5,8–10]. However, the key steps of MR biogenesis have been less explored. The sequence of events leading to the formation of functional protein should comprise at least insertion

[1]Institute of Biochemistry and Molecular Medicine, University of Bern, 3012 Bern, Switzerland. [2]Present address: Department of Biochemistry, University of Oxford, OX1 3QU, Oxford, UK. ✉e-mail: stephan.hirschi@bioch.ox.ac.uk; thomas.lemmin@unibe.ch; dimitrios.fotiadis@unibe.ch

of the protein into the membrane, protein folding, oligomerisation and binding of the retinal cofactor. GPR possesses an N-terminal signal sequence, akin to numerous membrane proteins, that is hypothesised to guide the correct insertion of the protein into the plasma membrane before subsequent proteolytic removal[11]. Interestingly, a recent large-scale metagenomics analysis observed that proteorhodopsin genes and genes necessary for retinal biosynthesis exhibit slightly diverging abundances, especially in deeper waters[12]. Another study that used a novel functional metagenomic screen discovered clones of marine SAR86 bacteria and freshwater actinobacteria lacking any recognisable pathways for the biosynthesis of retinal[13]. Together, these findings suggest that certain proteorhodopsin-expressing bacteria may not synthesise retinal endogenously, instead relying on scavenging the essential pigments from the surrounding environment, such as from sinking biomass. This would require the protein to have a transiently stable apo-state before incorporating the cofactor to yield functional proteorhodopsin. Evidence for the existence of a stable apoprotein was provided by experiments that demonstrated that chemical bleaching of proteorhodopsin with hydroxylamine could be reversed by the addition of all-*trans* retinal and that functional protein was formed by addition of the cofactor (or analogues) to membranes containing proteorhodopsin expressed in the absence of all-*trans* retinal[8,14].

For structural and functional characterisation, as well as for biotechnological applications, GPR is exclusively overexpressed in heterologous expression hosts such as *Escherichia coli*. However, this approach faces two main challenges related to GPR biogenesis: (i) the incomplete cleavage of the N-terminal signal sequence after membrane insertion in *E. coli* and (ii) the lack of endogenous retinal cofactor synthesis. The inhomogeneity resulting from incomplete cleavage of the signal peptide can be prevented by genetically removing it[15–17]. Conflicting results in the literature suggest that the oligomeric assembly of GPR can be either pentameric, hexameric or a mixture of both[15,18–21]. Notably, all studies reporting evidence for hexameric GPR used a construct containing the N-terminal signal sequence[15,19–27]. In contrast, the utilisation of an N-terminally truncated variant resulted in the exclusive observation of the pentameric form by cryo-EM[10,18]. This might indicate that the presence of the signal peptide is related to the mixtures of oligomeric states observed in the literature. The second obstacle, which pertains to the lack of endogenous retinal synthesis pathways in heterologous expression hosts, is routinely circumvented by supplementation of all-*trans* retinal during overexpression or by genetically introducing enzymes for a retinal biosynthesis pathway[28]. However, no structural information is available on the apoprotein and how externally supplied retinal is incorporated.

In this work, we elucidate the structural basis of distinct stages in the biogenesis of proteorhodopsins. We solve the cryo-EM structures of pentameric and hexameric GPR, revealing the effect of the N-terminal signal sequence on GPR oligomerisation. Furthermore, we determine the structure of the previously elusive retinal-free apo-state, proteoopsin (PO), thereby uncovering its significance for retinal scavenging. We observe the elongated density of an unknown ligand in the chromophore binding pocket of PO and identify it by mass spectrometry as decanoate. Using molecular dynamics (MD) simulations, we investigate the role of decanoate and other related compounds as temporary placeholders for retinal, potentially facilitating cofactor scavenging from the environment. Finally, through additional MD simulations, we explore possible pathways for the exchange of molecules in the chromophore binding pocket and propose a mechanism for retinal scavenging. In summary, our results hold significance for the large family of MRs and how endogenously synthesised or scavenged retinal is incorporated in these related proteins.

## Results and Discussion
### Overall structures of pentameric and hexameric GPR-A18L variants

To investigate the mixture of pentameric and hexameric GPR observed in the literature, we assessed whether the resulting oligomeric assembly depends on the presence of the N-terminal signal peptide. To this end, we expressed the GPR-A18L variant, introducing an alanine to leucine substitution (A18L) known to prevent cleavage of the signal peptide in the blue-light absorbing proteorhodopsin (BPR) when expressed in *E. coli*[11]. Purification of the GPR-A18L variant resulted in a pure sample that exhibited a higher molecular weight (Supplementary Fig. 1A) than the processed wild-type protein[16]. Furthermore, no partial processing of the N-terminal signal sequence was observed, as evidenced by a single band, in contrast to a double band commonly found in wild-type GPR[16]. We thus demonstrated that the introduction of the A18L mutation in GPR effectively confers resistance against proteolytic cleavage of the N-terminal signal peptide when the protein was expressed in *E. coli*, similar to BPR[11]. Furthermore, the presence of the signal peptide in purified GPR does not impair oligomer formation as seen by size-exclusion chromatography (SEC) (Supplementary Fig. 1). Subsequent analysis of GPR-A18L oligomers by single particle cryo-EM revealed pentameric and hexameric particles. The dataset comprised ~710,000 pentamers and ~240,000 hexamers (Supplementary Fig. 2), agreeing with the observed stronger protein band intensity for pentamers compared to hexamers in blue native-polyacrylamide gel electrophoresis (BN-PAGE) (Supplementary Fig. 3A). The oligomer particle heterogeneity observed during 2D classification could be resolved using heterogeneous refinement in cryoSPARC[29], yielding a pentameric and a hexameric GPR 3D density map (Supplementary Fig. 2). Upon careful examination of the hexameric GPR map, two α-helical densities were identified in the central cavity, challenging the initially assumed $C_6$ symmetry and revealing a $C_2$ symmetry instead. No corresponding densities were found in the pentameric GPR map. By unravelling the two sets of particles, the structures of pentameric and hexameric GPR-A18L could be solved from a single data set to resolutions of 2.82 Å for the $C_5$ pentamer (PDB ID: 8CQC) and 3.54 Å for the $C_2$ hexamer (PDB ID: 8CQD) (Fig. 1).

The overall architecture of the GPR-A18L pentamer is practically identical to the previously solved N-terminally truncated GPR structure (PDB ID: 7B03)[10] with a $C_\alpha$ root-mean-square deviation (RMSD) of 0.33 Å. Only minor differences in the position of the amino acid side chains were observed (RMSD: 0.57 Å). No density for the N-terminal signal peptides could be found, presumably due to significant flexibility and a lack of interactions. In contrast, two α-helical densities were discovered filling the central cavity of the GPR-A18L hexamer, corresponding to the N-termini of protomers 1 and 4 (Fig. 1B). Accommodation of additional signal peptides in the cavity is restricted due to their tilt of ~30° relative to the membrane normal. The absence of densities for the remaining four N-termini indicates that they are likely located at the periphery of the oligomeric assembly. The fact that a mixture of pentamers and hexamers was observed, but that the signal peptides were only found in the central cavity of the hexamer suggests the assembly of GPR oligomers to be a stochastic process. The position of the N-terminal signal peptide relative to the heptahelical bundle during oligomer assembly might determine, which oligomer type is formed. If two signal peptides occupy the space that will form the central cavity of the oligomer, this expands the diameter and thus permits the assembly of six protomers. Conversely, if the signal peptides have moved away from the centre, the protomers can come into closer contact and form pentamers. Since the original γ-proteobacterial host is not available for culturing, it is not possible to investigate to what extent the signal peptide is processed naturally. Therefore, hexameric GPR could either be an artefact of recombinant expression in *E. coli*, or incomplete cleavage of the signal peptide

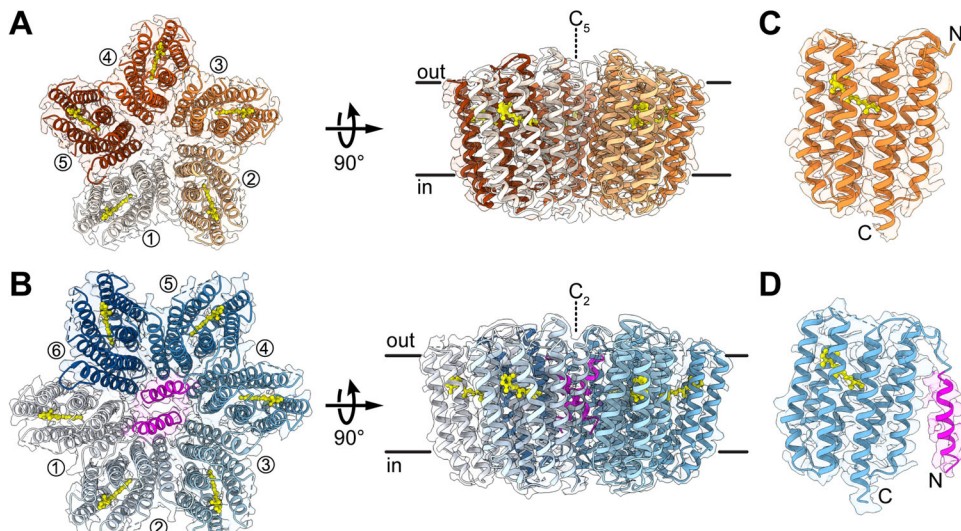

**Fig. 1 | Cryo-EM structures of GPR-A18L pentamer and hexamer.** Top views from the extracellular space and side views of GPR-A18L pentamer (**A**) and hexamer (**B**) structures. Individual protomers are labelled in the top views, lipid bilayer boundaries (based on the PPM server[64]) and symmetry axes are indicated in the side views. Interacting *N*-terminal signal peptides of protomers 1 and 4 in the GPR-A18L hexamer are highlighted in magenta. Close-up views of single protomers from pentameric (**C**) and hexameric (**D**) GPR-A18L with N- and C-termini indicated are shown. The α-helical structure of the N-terminal signal peptide is highlighted in magenta and the retinal cofactor is represented by a ball-and-stick model (yellow).

represents a way to regulate formation of hexamers and pentamers in the native host.

## Comparison of interprotomer interactions, spectral properties and thermostability between pentameric and hexameric GPR-A18L

In the GPR-A18L hexamer, the signal peptides of protomers 1 and 4 not only directly interact with each other, but also engage the other protomers (Fig. 2A). Due to the $C_2$ symmetry of the hexameric structure, the interactions between the two symmetry-related protomer groups are identical. We will therefore focus our discussion on protomers 1-4. The interface between protomers 1 and 2 is extended by hydrophobic interactions of the signal peptide with TMHs A and B of protomer 2. Furthermore, hydrogen bonds are formed between Y79 in protomer 2 and the carbonyl oxygens of the N-terminal P15 and L18 amino acids (Fig. 2B). A minor interface is formed between the tip of the signal peptide of protomer 1 and TMH A of protomer 3 (Fig. 2C). The interfaces between protomers 1 and 4 are mainly formed by hydrophobic interactions where the signal peptides intersect and where they interact with TMH A of the opposing protomer (Fig. 2D). This includes a hydrogen bond formed between the two serine residues at position 10 and a salt bridge between K3 and D53 on TMH A (Fig. 2D). In total, these additional interactions, formed by the signal peptides with neighbouring protomers, account for an increased oligomerisation interface area compared to the pentamer of around 1400 Å² (about 25% more), as calculated by PISA[30]. Interestingly, protomers 1 and 4 exhibit slight differences compared to the other protomers (Supplementary Fig. 4A), likely due to the interaction of their signal peptides in the centre of the hexamer. This conformational change mainly affects the extracellular halves of TMHs E and F, and is also noticeable in the position of the retinal chromophore (Supplementary Fig. 4A). In contrast, the structures of protomers 2, 3, 5 and 6 in the hexamer are all very similar (Supplementary Fig. 4B), and share a comparable structure with the protomers of the GPR-A18L pentamer (Supplementary Fig. 4C).

The slightly altered conformation of protomers 1 and 4 compared to the others in hexameric GPR and the resulting subtle change in retinal conformation (Supplementary Fig. 4A) raised the question if these differences affect the spectral properties of the hexamer compared to the pentamer. Since GPR pentamers and hexamers cannot be resolved by SEC, we separated them by BN-PAGE (Supplementary Fig. 3A) followed by protein extraction from the gel and removal of the Coomassie dye. Isolated pentamers and hexamers were then analysed by UV-Vis spectroscopy, revealing absorption spectra with comparable maxima for the retinal Schiff base, thereby demonstrating similar spectral properties for the two oligomeric forms (Supplementary Fig. 3B).

Another intriguing consideration is whether the hexameric assembly affects interprotomer interactions compared to pentameric GPR. Analysis of the oligomerisation interfaces between neighbouring protomers revealed comparable interactions between residues (Supplementary Fig. 5A), which were already discussed in detail for the pentamer previously[10]. In both the GPR-A18L pentamer and hexamer, the same two interaction areas are observed, comprising a large hydrophobic patch running along the membrane-embedded region and a polar interaction network towards the intracellular end. While most interactions involve the same residues, the hexameric GPR interface is slightly larger due to the closer proximity of the protomers and the narrower angle of encounter. This results in a larger surface area of 950 Å² compared to 790 Å² in the pentamer (Supplementary Fig. 5B), excluding interfaces involving the signal peptide. Despite this increased contact area, nanoDSF analysis of isolated GPR-A18L pentamers and hexamers indicates greater stability of the pentamer (Supplementary Fig. 3C). This discrepancy might be attributed to suboptimal packing at the hexamer interface, leading to weaker interactions.

A previous study has made an effort to correlate certain structural motifs with the propensity of MRs to form different oligomers[31]. Bacteriorhodopsin and related trimeric MRs feature shorter TMHs at their interfaces compared to the more extended helices found in Gloeobacter rhodopsin-like proteins, which exclusively form pentamers, and proteorhodopsins, which form both pentamers and hexamers. Whereas bacteriorhodopsin-like proteins feature a β-turn in the B-C loop facing towards the centre of the protomers, Gloeobacter rhodopsin-like proteins exhibit a 3-omega motif, a stack of three aromatic residues between TMHs A and B, which causes the B-C loop to flip away from the centre. In contrast, proteorhodopsins have very short B-C loops and are missing the β-turn structure completely. The

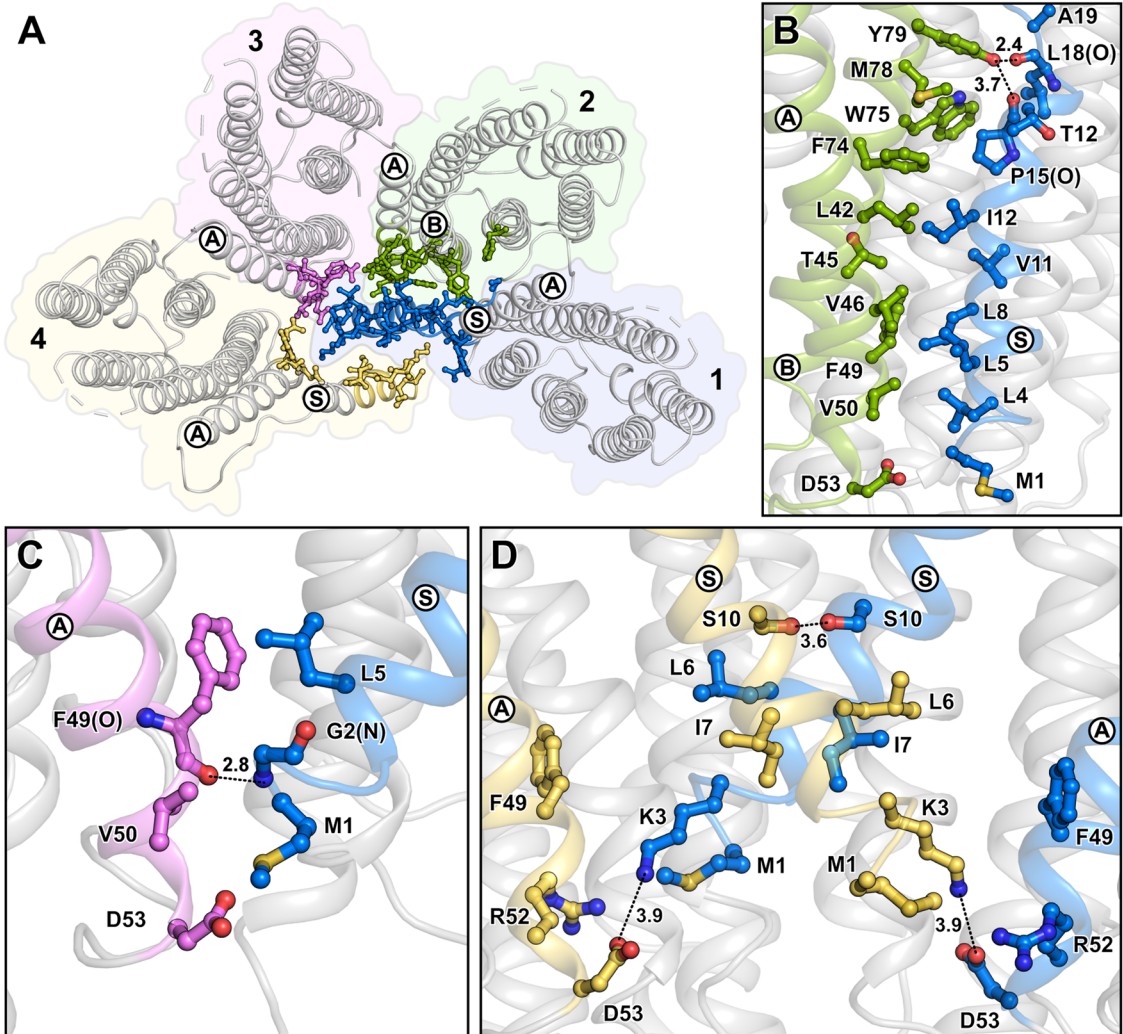

**Fig. 2 | Oligomerisation interfaces and interactions involving the N-terminal signal peptide in hexameric GPR-A18L. A** Overview of interfaces formed between protomers 1–4 (coloured individually) involving the signal peptides of protomers 1 and 4 with interface-forming amino acids highlighted. Protomers 5 and 6 form identical interactions with protomers 1 and 4, and were therefore omitted for clarity. Detailed view of amino acid interactions making up the interfaces between protomers 1 and 2 (**B**), 1 and 3 (**C**) and 1 and 4 (**D**). Helices bearing amino acids participating in these interfaces are highlighted in the corresponding colour of the protomers and are labelled A, B or S (signal peptide). Hydrogen bond distances (dotted lines) are indicated in Å.

presented structures of pentameric and hexameric GPR-A18L confirm these observations and agree with the categorisation based on these structural features. Furthermore, our findings reveal how the unprocessed signal peptide contributes to the formation of the hexamer but not the pentamer, and how the presence of this additional N-terminal α-helix affects the structure of individual protomers and the interfaces between them.

**Structure and LC-MS of proteoopsin reveal decanoate as temporary placeholder for retinal**

After having gained a deeper understanding of the variations in the oligomeric assembly of GPR and their consequences, we focused on the incorporation of the retinal cofactor, a crucial step in the biogenesis of all microbial rhodopsins. Intriguingly, some proteorhodopsin-expressing bacteria were found to lack the means to synthesise their own retinal, which suggests the existence of a stable apoprotein[12,13]. To characterise this retinal-free state, hereafter referred to as proteoopsin (PO), we successfully expressed and purified an N-terminally truncated GPR version, lacking the signal peptide, without the addition of all-*trans* retinal (Supplementary Fig. 1). The oligomeric assembly of PO was confirmed by SEC and the lack of absorption around 525 nm

demonstrated the absence of the retinal Schiff base (Supplementary Fig. 1C) These results collectively indicate that retinal-free PO is stable and oligomer formation is independent of chromophore binding. Using single particle cryo-EM, the structure of PO was solved at a resolution of 2.97 Å (Supplementary Fig. 6; PDB ID: 8CNK). PO exhibits a pentameric assembly (Fig. 3A) with an overall structure similar to that of GPR[10] (PDB ID: 7B03; $C_\alpha$ RMSD of 1.65 Å) and the GPR-A18L pentamer (Fig. 1A; $C_\alpha$ RMSD of 1.63 Å), forming the same interface interactions. This suggests that the absence of retinal does not significantly affect the overall oligomeric assembly of PO.

Closer inspection of the retinal binding pocket revealed an elongated density, corresponding to an unknown ligand (Fig. 3A). The dimensions of this density are comparable to all-*trans* retinal and it occupies the same position within the pocket. However, it is more slender and uniform compared to all-*trans* retinal, lacking the bulky density associated with the β-ionone ring. Additionally, the density is not connected to K232, indicating a possible non-covalent interaction. The molecule was extracted from purified PO with hydroxylamine, a common and efficient extraction method for retinal, and analysed using the first quadrupole mass filter (Q1) in electrospray ionisation (ESI) tandem mass spectrometry (MS/MS). From these scans (20-

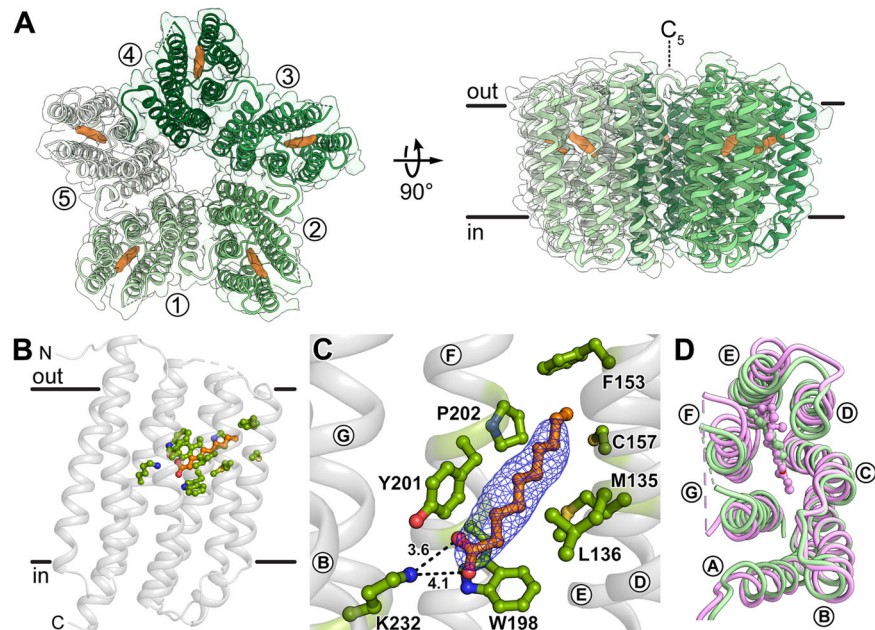

**Fig. 3 | Cryo-EM structure of retinal-free proteoopsin (PO). A** Top and side views of the PO pentamer cryo-EM structure. Individual protomers are labelled in the top view (seen from the extracellular side), lipid bilayer boundaries (based on the PPM server[64]) and symmetry axis are indicated in the side view. Ligand densities are highlighted in orange. **B** The decanoate molecule (orange) in the chromophore binding pocket is shown with interacting residues (green). **C** Close-up view of the chromophore binding pocket with decanoate cryo-EM density (blue mesh). The carboxylate of decanoate forms a salt bridge with K232 (interaction distances indicated in Å) and the acyl chain is surrounded by hydrophobic residues. **D** Structural comparison of decanoate-bound PO (green) and retinal-bound GPR (magenta, PDB ID: 7B03) viewed from the extracellular space. Transmembrane α-helices are labelled (A–G) in (**C**) and (**D**).

200 Da range, i.e., for molecular masses of potential ligands fitting into the identified cryo-EM density), fragments of the precursor ions (Q2) were determined (Supplementary Fig. 7). The experiment was validated using retinal-bound GPR lacking the N-terminal signal sequence (GPR$_{\Delta18}$), i.e., only signals found in PO but not in GPR$_{\Delta18}$ were considered for the ligand in question. Major molecular ion peaks and fragments consistently observed in multiple independent Q1 and Q2 ESI-MS/MS experiments were matched against spectral databases LipidBlast (fiehnlab.ucdavis.edu/projects/lipidblast) and Metlin (metlin.scripps.edu). LC-MS scans from the enhanced product ion (EPI) mass spectra of *N*-hydroxydecanamide showed the specific ions at 188 *m/z*, 118 *m/z* and 104 *m/z* (recombined ions) (Supplementary Fig. 7). This led to the identification of the unknown ligand as decanoic acid (capric acid) based on the detection of its hydroxylamine solvent reaction product *N*-hydroxydecanamide. Noteworthy, neither retinal nor decanoic acid could be extracted using common lipid extraction methods, like the Folch method or using ethyl acetate plus hexane (9:1) and 0.1% (v/v) formic acid. However, decanoic acid could only be extracted when using hydroxylamine, as the solvent reaction product *N*-hydroxydecanamide. This can be attributed to the particular denaturation properties of hydroxylamine, known to destabilize microbial rhodopsins, including the highly thermostable bacteriorhodopsin[32,33]. This allowed efficient extraction of retinal, and in our case, also decanoic acid, from the rhodopsins, which are tightly packed around the cofactor. The fact that only hydroxylamine was able to extract decanoic acid from PO, further substantiates that it is tightly bound in the retinal binding pocket and not just peripherally associated.

The presence of the reaction product *N*-hydroxydecanamide, and thus decanoic acid as endogenous ligand, in the PO sample was subsequently confirmed and quantified employing a liquid chromatography (LC)-ESI-MS/MS multiple reaction monitoring (MRM) method (Supplementary Fig. 8). Using LC-MS/MS, *N*-hydroxydecanamide was found only in the PO samples and could not be detected in the GPR$_{\Delta18}$ samples (Supplementary Fig. 8). Moreover, a

semiquantitative estimation of total PO protein and decanoate in samples yielded close to equimolar concentrations, in agreement with the assumption that there is one ligand per PO protomer. Consequently, one decanoate molecule (p$K_a$ in water is 4.9) per protomer was built into the cryo-EM density.

Decanoate is non-covalently bound to PO, with its carboxylate group pointing towards the GPR Schiff base region and its acyl chain extending towards the extracellular end of TMH E (Fig. 3B). Instead of the retinal Schiff base, a salt bridge is formed through the engagement of the negatively charged carboxylate with the side chain of K232 (Fig. 3C). When comparing the decanoate- and retinal-bound structures, the spatial orientation of the two ligands within the binding pocket, as well as their interactions with amino acid side chains in the hydrophobic part of the pocket, are mostly identical due to their similar lengths (Fig. 3D and Supplementary Fig. 9). However, significant shifts were observed in the extracellular halves of TMHs D-F (Fig. 3D). This conformational difference can likely be ascribed to the bulkiness of the β-ionone ring, which occupies more space than the slender acyl chain of decanoate. Overall, similar dimensions to retinal and the interaction of its carboxylate group to K232 make decanoate a suitable temporary placeholder for retinal during the biogenesis of GPR when the chromophore is not available.

**Decanoate affects proteoopsin dynamics and modulates binding pocket accessibility**

To investigate the effect of decanoate and retinal on the conformational dynamics of PO, we conducted MD simulations with the structure of PO and the previously published GPR structure (PDB ID: 7B03)[10]. We compared simulations of both proteins with all-*trans* retinal bound by a Schiff base, non-covalently bound decanoate and no ligand (Fig. 4, Supplementary Figs. 10 and 11). All simulations were carried out with the proteins embedded in a lipid bilayer mimicking a bacterial membrane to simulate near-physiological conditions. The retinal-bound proteins sampled a relatively confined conformational space, indicative of a stable structure, and converged around the

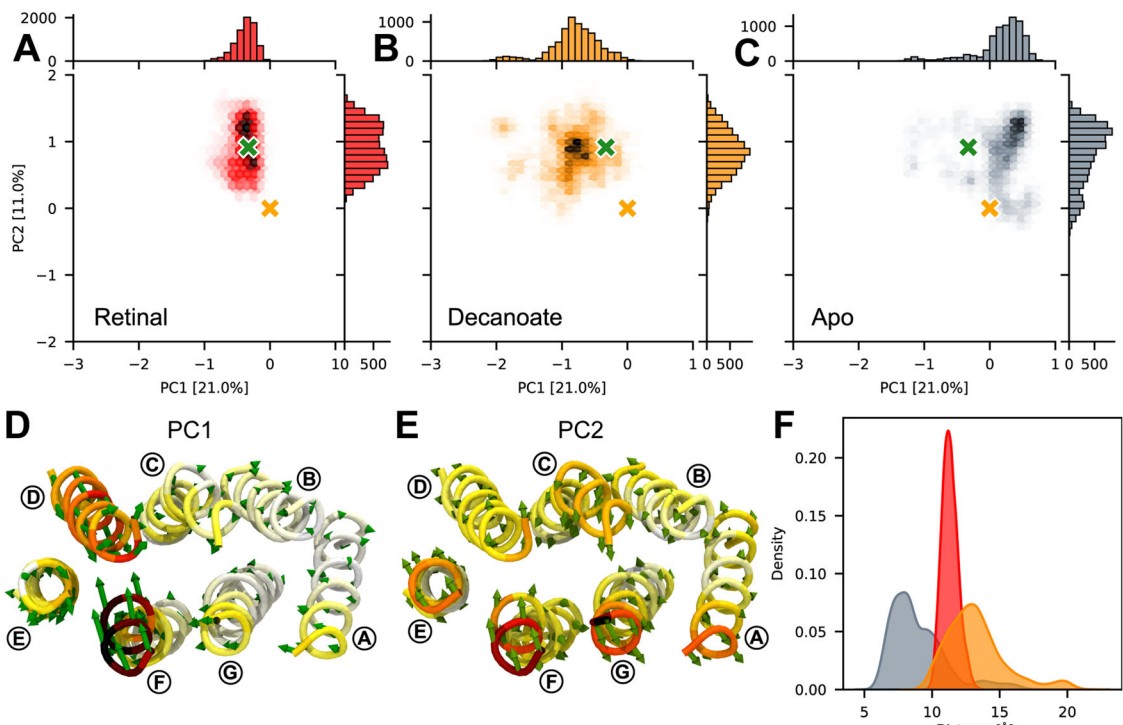

**Fig. 4 | Molecular dynamics (MD) simulations of retinal- or decanoate-bound, and ligand-free proteoopsin (PO).** Shared eigenspace defined by principal component (PC) analysis on aggregated MD simulations starting from lipid-embedded structures of PO (orange cross) or green-light absorbing proteorhodopsin (PDB ID: 7B03; green cross) bound to all-*trans* retinal by Schiff base (**A**), bound non-covalently to decanoate (**B**) or without ligand (apo; **C**). The conformational densities are represented by hexagons, with a higher colour intensity reflecting a larger number of states in each bin. Marginal distributions for each PC are displayed on the respective axes. Top view of PO structure (seen from extracellular side) with eigenvectors corresponding to PC1 (**D**) and PC2 (**E**) observed during the simulations represented as green arrows. The model is coloured to reflect the magnitude of the displacement, from small (white) to larger movement (red). Transmembrane α-helices are labelled (A–G) and loops were omitted for clarity. **F** Distribution of observed distances between residues L136 and P202 during the simulations to assess opening of the binding pocket. The same colours are used in (**A–C**) and (**F**) to represent the simulations of retinal-bound (red), decanoate-bound (orange) and ligand-free PO (apo; grey).

original retinal-bound GPR structure (Fig. 4A). In contrast, simulations with decanoate-bound proteins were found to be more dynamic, covering a wider range of conformations (Fig. 4B). In particular, the extracellular halves of TMHs D and F, and to a lesser extent TMH E, exhibited significant mobility, resulting in an expansion of the binding pocket (Fig. 4B and D). This conformational flexibility might serve as a mechanism for facilitating the substitution of decanoate with retinal, allowing the protein to scavenge the cofactor from the environment. Without a ligand occupying the binding pocket, the extracellular half of the apoprotein partially closed, with the top part of TMH F moving towards the centre (Fig. 4C). To quantify the extent to which the binding pocket opens or closes, we measured the distance between residues L136 and P202. These residues were identified as key residues exhibiting significant movement in the principal components and directly flank the β-ionone ring of retinal in the GPR structure. In the simulations without ligand, the distance between residues L136 and P202 significantly decreased, indicating a partial closing of the binding pocket (Fig. 4F). Conversely, in the simulations with decanoate the distance between these residues increased, reflecting an opening of the binding pocket (Fig. 4F). This suggests that when decanoate is bound to PO, the protein adopts a transitional state with facilitated access to the binding pocket, until retinal is available and incorporated. The role of decanoate as a placeholder is further supported by its high binding affinity to PO, as evidenced by its ability to co-purify with the protein.

Since PO was heterologously expressed in *E. coli*, it is important to consider that decanoate may not naturally occur in proteorhodopsin-expressing hosts and that this placeholder role may be fulfilled by a molecule with comparable properties. Through a series of MD simulations, we evaluated the suitability of a selection of chemically related compounds as potential substitutes for decanoate. A total of nine different molecules were tested, featuring acyl chains of varying length (8, 10 or 12 carbon atoms) and head groups with different oxidation states (alkanes, primary alcohols and carboxylates). The choice of molecules to be tested was based on the spatial constraints posed by the protein and potential availability in the membrane, such as intermediates from the fatty acid metabolism. We analysed the correlation between the simulated ligand density derived from the MD simulations and the experimental cryo-EM density. This provides a good estimation for spatial requirements and the behaviour of the ligands in the binding pocket (Supplementary Fig. 12). The highest correlation was found for polar ligands, i.e., alcohols and carboxylates, with chain lengths of 10 and 12 carbon atoms. These findings suggest that medium-chain alcohols and carboxylates may potentially serve as temporary placeholders for retinal in organisms that do not synthesise retinal.

## Potential pathways for retinal scavenging and chromophore formation

In the absence of all-*trans* retinal, the chromophore binding pocket of PO was found to be occupied by decanoate, or potentially similar molecules in native hosts. The functionality of decanoate as a placeholder in the absence of retinal was supported by the visual formation of the retinal Schiff base upon addition of free all-*trans* retinal to detergent-solubilised PO (Supplementary Fig. 13). This confirmed that retinal can effectively replace decanoate in the binding pocket and

restore the functional state of the protein. Additionally, MD simulations showed that the structure of PO is more dynamic, further indicating the necessity of a molecule in the binding pocket to prevent collapse and maintain accessibility until retinal becomes available for incorporation (Fig. 4).

Interestingly, the chromophore binding pocket in PO exhibits a narrow opening between the extracellular halves of TMHs E and F, which could potentially provide access for decanoate and retinal (Supplementary Fig. 14A). Similar openings were also found in the ligand-free structures of the G protein-coupled receptor opsin and the channelrhodopsin ChRmine (apo-ChRmine) (Supplementary Fig. 14B and C)[34,35]. While these openings do not extend beyond the binding pocket in PO and apo-ChRmine, in opsin it is part of a channel with a second exit that was proposed to allow the exchange of all-*trans* and 11-*cis* retinal molecules[35–37]. Importantly, the mechanisms of retinal access might be distinct for each protein, with PO and apo-ChRmine featuring more of a cleft and opsin a continuous channel. To investigate the entry and exit pathways of decanoate and retinal in PO, MD simulations were conducted on both ligand-bound structures. To explore potential exit pathways, random acceleration MD (RAMD) simulations were performed, applying a force with random orientation to the centre of mass of each ligand. These simulations yielded multiple trajectories, which were grouped into five potential exit pathways shared by retinal and decanoate, and an additional distinct pathway exclusively observed in the decanoate simulations (Fig. 5). In pathways 1 (cyan) and 2 (green), both ligands exit PO towards the extracellular space near TMHs A or E, respectively, and seem to diffuse into the aqueous bulk solution. The third group of trajectories (yellow) shows the molecules moving between TMHs C and D, with retinal trailing off towards the extracellular space while decanoate exits closer to the centre of the lipid bilayer, mostly remaining in the membrane. Following pathways 4 (blue) and 5 (magenta), the molecules exit either between TMHs D and E, or between TMHs E and F, respectively. In both

cases, most trajectories remain within the lipid bilayer. Notably, a distinct exit pathway (6) was observed for decanoate between TMHs F and G that, in most cases, leads into the extracellular space.

The octanol-water partition coefficients (LogP) for decanoate and retinal were considered to assess the likelihood of the different pathways. The LogP for decanoate is 4.0, which translates to a predicted water solubility of about 280 µM, and the LogP for retinal is 7.6, with a predicted water solubility of 0.017 µM (values taken from ChemSpider, calculated using KOWWIN v1.67). As expected, the negative charge of the carboxyl group endows decanoate with higher water solubility compared to retinal, potentially opening up different pathways. Considering the overall hydrophobic nature of both ligands, pathways 1, 2 and 3 are unfavourable exit pathways for retinal, and pathways 1, 2 and 6 are unfavourable for decanoate, since these pathways all lead towards the aqueous environment (Fig. 5). Given the high flexibility exhibited by TMHs E and F during MD simulations of decanoate-bound PO (Fig. 4) as well as the observed opening similar to opsin and apo-ChRmine (Supplementary Fig. 14), trajectory 5 appears to be the most likely pathway into the lipid bilayer for the exchange of decanoate and retinal. Decanoate could potentially also access the protein through the hydrophobic pathways 3 and 4, whereas pathway 4 could represent an alternative entry point for retinal (Fig. 5). Further experimental validation will be required to confirm a definitive mechanism, but it is possible that both molecules are able to use more than one pathway and that they may not use the same one to enter and exit the protein.

## Integrative insights into the key steps of proteorhodopsin biogenesis

This study provides detailed insights into the structure and dynamics of key steps in the biogenesis of proteorhodopsin, including: (i) elucidating the role of the signal peptide in the heterogeneous oligomerisation of GPR, (ii) uncovering the presence of decanoate as a temporary placeholder in retinal-free PO, (iii) exploring the potential

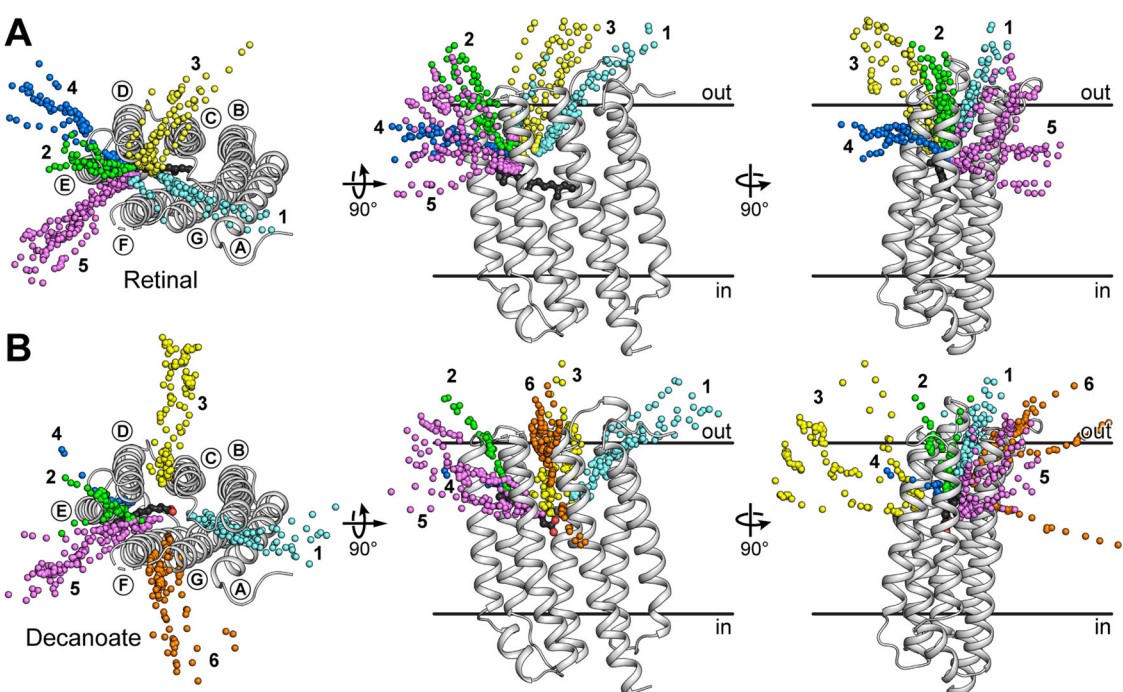

**Fig. 5 | Random acceleration molecular dynamics (RAMD) simulations exploring potential exit pathways of non-covalently bound all-*trans* retinal and decanoate from proteoopsin.** The exit trajectories for the non-covalently bound all-*trans* retinal (**A**) or decanoate (**B**) are shown with spheres representing the centres of mass of the ligands. Similar trajectories were organised into groups labelled 1-6 (coloured individually), where pathway 6 was only observed in decanoate simulations. Top views in (**A**) and (**B**) are seen from the extracellular side. Ligands are depicted as ball-and-stick models (black), TMHs are labelled (A–G) and lipid bilayer boundaries (based on the PPM server[64]) are indicated in the side views.

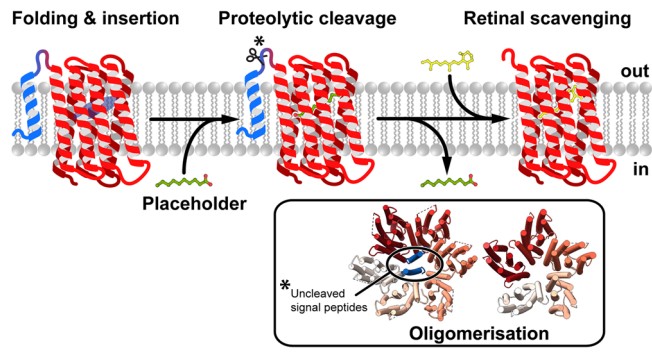

**Fig. 6 | Schematic model of major steps in proteorhodopsin biogenesis.** After the nascent peptide is integrated into the lipid bilayer and has folded into the characteristic seven α-helical transmembrane bundle, the structure is stabilised by decanoate, or other similar endogenous placeholder molecules if all-*trans* retinal is not available. When the N-terminal signal peptide (blue) is not removed, a mixture of hexamers and pentamers are observed. Finally, the placeholder molecule is displaced by all-*trans* retinal scavenged from the environment to form functional GPR.

of related molecules as placeholders and (iv) identifying possible pathways for the exchange of decanoate and retinal, allowing the scavenging of the essential cofactor from the environment (Fig. 6). Numerous studies have addressed GPR structure-function relationships, some demonstrating the effect of oligomerisation on photocycle kinetics and proton transport[23,25,27]. However, the reason for the existence of a mixture of pentameric and hexameric oligomers remained unclear. Using a GPR mutant with an uncleavable signal peptide, we solved the cryo-EM structures of pentameric and hexameric GPR, revealing that the signal peptide is involved in the formation of the GPR hexamer but not the pentamer. This provides a plausible explanation for the mixed oligomeric states observed in heterologously expressed GPR samples, where the signal peptide is only partially processed and thus might facilitate the formation of hexamers in a stochastic assembly process. Thus, if homogeneous GPR samples are required for biochemical or biophysical studies, constructs lacking the signal peptide should be considered. Since no evidence is available on how the signal peptide is cleaved in native hosts, two main scenarios are conceivable: (i) the hexamer might either be an artefact of recombinant expression with the pentamer representing the natural oligomeric state or (ii) incomplete cleavage of the signal peptide occurs in the native hosts and represents a way to regulate the formation of hexamers and pentamers.

To yield functional proteorhodopsin or, by extension, any functional retinal-dependant microbial rhodopsin, all-*trans* retinal needs to be incorporated by forming a Schiff base with a central lysine in the chromophore binding pocket. We demonstrated that retinal-free PO can be expressed and successfully purified, indicating that the protein is stable even in the absence of the chromophore. Solving the cryo-EM structure of PO combined with MS experiments and MD simulations revealed the presence of a decanoate molecule in the chromophore binding pocket, which we propose to act as a temporary placeholder for retinal. Furthermore, using MD simulations we have shown that depending on the availability in the original organisms, similar molecules to decanoate, including C10-12 alcohols and carboxylates, could also act as placeholders in the apoprotein (Supplementary Fig. 12). Carboxylic acids with an even number of carbon atoms are intermediates in the bacterial fatty acid biosynthesis, which provides the most likely source for these molecules. Using RAMD simulations, we explored pathways for the exit of decanoate and retinal from the protein, offering a potential mechanism by which the two molecules can be exchanged. One of the two most likely pathways, leading between TMHs E and F (trajectory 5 in Fig. 5), resembles an opening in

apo-ChRmine and the proposed retinal channel in opsin[34,35]. In vertebrates, the G protein-coupled receptor rhodopsin goes through cycles of binding 11-*cis* retinal and releasing all-*trans* retinal, whereas retinal remains covalently bound throughout the photocycle in microbial rhodopsins. Given the structural similarity of opsin with GPR, the retinal uptake mechanism might be, at least partially, conserved between microbial and mammalian rhodopsins. With examples of organisms lacking a retinal synthesis pathway from freshwater and salt water, insight into how retinal can be acquired and incorporated potentially carries broader relevance for all microbial rhodopsins.

Collectively, our findings shed light on the intricate mechanisms underlying proteorhodopsin biogenesis, including oligomer assembly and the effect of the signal peptide on protomer stoichiometry, the role of a ligand as a placeholder in the functional maturation of the apoprotein and finally a potential mechanism for retinal scavenging from the environment. These insights advance our understanding of proteorhodopsin's structural and functional dynamics and highlight critical areas for further investigation in microbial rhodopsin research.

## Methods

### Cloning and overexpression of proteorhodopsin versions

The wild-type GPR gene, with or without (GPR$_{\Delta18}$) the N-terminal signal sequence (MGKLLLILGSVIALPTFA), was cloned into the pZUDF21-5H vector, encoding for a C-terminal penta-histidine tag. The QuikChange Lightning Multi Site-Directed Mutagenesis Kit (Agilent) was used to generate the GPR-A18L mutant, yielding a version with an uncleavable signal sequence. Both constructs were expressed in *Escherichia coli* BL21 (DE3) Rosetta 2 as described previously[17]. After growing the cells at 37 °C to an OD$_{600}$ of 0.75, expression was induced by the addition of 0.1 mM isopropyl-β-D-thiogalactopyranoside (IPTG) and 5 μM all-*trans* retinal (100 mM stock solution in 100% ethanol). For the expression of proteoopsin, no all-*trans* retinal was added. After induction, the temperature was reduced to 20 °C and cells were grown overnight. Cells were washed once in 50 mM Tris-HCl pH 8, 450 mM NaCl by centrifugation at 10,000 x g and 4 °C for 10 min, resuspended in the same buffer and stored at -20 °C until further use.

### Purification of GPR$_{\Delta18}$, GPR-A18L and PO

Bacterial membranes containing the overexpressed proteorhodopsin variants were isolated as described previously[17]. Briefly, cells were thawed and lysed using a Microfluidizer (M-110P Microfluidizer, Microfluidics) at 1,500 bar for five cycles. Cell debris was removed by centrifugation at 10,000 x g and 4 °C for 10 min (Optima L-90K ultracentrifuge, Beckman Coulter). Membranes were collected by ultracentrifugation at 150,000 x g and 4 °C for 1 h. The pellets were homogenised in 50 mM Tris-HCl pH 8, 450 mM NaCl, washed twice and finally homogenised in solubilisation buffer (20 mM 4-(2-hydroxyethyl)piperazine-1-ethanesulfonic acid (HEPES)-NaOH pH 7.5, 300 mM NaCl, 10% (v/v) glycerol).

An aliquot corresponding to membranes isolated from 3 L bacterial cell culture was solubilised overnight at 4 °C in a total volume of 21 mL solubilisation buffer containing 3% (w/v) 5-cyclohexyl-1-pentyl-β-D-maltoside (Cymal-5). Unsolubilised material was removed by ultracentrifugation at 100,000 x g and 4 °C for 1 h. The supernatant was diluted with 21 mL wash buffer (20 mM HEPES-NaOH pH 7.5, 300 mM NaCl, 60 mM imidazole) and bound to 1.5 mL equilibrated Ni-NTA Superflow resin (Qiagen) by repeated application on a gravity flow column (Promega Wizard Midi columns) for 3 h at room temperature. The column was washed three times with 7 mL wash buffer containing 0.25% (w/v) Cymal-5 before eluting the protein in the same buffer but with 400 mM imidazole. The protein was concentrated to 500 μL using a Vivaspin 2 mL ultrafiltration spin column with 50 kDa molecular weight cut-off (Sartorius) and applied to a Superdex 200 Increase 10/300 column mounted on an Äkta Purifier (Cytiva) and equilibrated with 20 mM Bis-Tris propane (BTP)-HCl pH 7.5, 150 mM NaCl,

0.25% (w/v) Cymal-5 (SEC buffer). SEC fractions containing oligomers (Supplementary Fig. 1) were collected and concentrated again for cryo-EM. Concentrated proteins were ultracentrifuged at 100,000 x g and 4 °C for 15 min prior to cryo-EM sample preparation.

For mass spectrometry analysis, PO and GPR$_{\Delta 18}$ were purified using the same procedure as described above, except that 200 mM L-histidine were used for elution from Ni-NTA resin and SEC was performed using a Superdex 200 Increase 3.2/300 column (Cytiva) equilibrated with SEC buffer containing 0.15% (w/v) Cymal-5. Peak oligomer fractions from SEC with a concentration of about 2 mg/mL were directly analysed.

## Cryo-EM sample preparation and data collection

3 µL of purified GPR-A18L at 4.2 mg/mL or PO at 4.0 mg/mL were applied to either Quantifoil R 2/1 200 mesh or R 1.2/1.3 300 mesh copper holey-carbon grids that were glow discharged for 60 s at 10 mA and 0.25 mbar (PELCO easiGlowTM system). Grids were blotted for either 2 s (GPR-A18L) or 3 s (PO) using a Vitrobot Mark IV (Thermo-Fisher) at 4 °C and 100% humidity, and immediately vitrified by plunging into liquid ethane. Data were collected on a Titan Krios G3 (ThermoFisher) operated at 300 kV and equipped with a Quantum-K3 direct electron detector (Gatan) using SerialEM[38]. For GPR-A18L, a total of 21,712 movies were recorded at a magnification of 105,000x with a calibrated pixel size of 0.822 Å and a defocus range of -0.7 to -1.7 µm. For PO a total of 20,910 movies were acquired at a magnification of 130,000 x with a calibrated pixel size of 0.645 Å and a defocus range of -0.7 to -1.7 µm. Movies of both data sets each comprised 40 frames with a total dose of 49.8 e$^-$/Å$^2$.

## Cryo-EM data processing

The cryo-EM data processing workflows for PR-A18L and PO are illustrated in Supplementary Figs. 2 and 6, respectively. Validation statistics are presented in Supplementary Table 1, particle orientations, local resolution and Fourier shell correlation (FSC) plots in Supplementary Fig. 15 and cryo-EM map quality in Supplementary Figs. 16 and 17. Patch-based motion correction of dose-fractionated and gain-corrected movies was performed in Relion 3.1.1[39] using MotionCor2[40] (version 1.4.0). Contrast transfer function (CTF) estimation was performed with ctffind 4.1.14[41] and used to reject micrographs with abnormal defocus, astigmatism and low figure of merit. Finally, micrographs with a maximum CTF resolution better than 6.0 Å (GPR-A18L) or 5.5 Å (PO) were selected, resulting in a total of 18,017 and 18,044 micrographs, respectively. An initial particle set was picked from a subset of micrographs using the Laplacian-of-Gaussian (LoG) filter in Relion to generate templates for autopicking. Using selected 2D classes as templates, 4,382,612 particles were picked from the GPR-A18L data set and 4,048,441 particles from the PO data set.

GPR-A18L particles were directly exported to cryoSPARC after picking. Curation by two rounds of 2D classification yielded a set of 1,418,744 particles and revealed pentameric and hexameric top views. Two ab initio models were generated and used in a subsequent heterogeneous refinement to separate particles with different oligomeric states. This yielded a map at moderate resolution and a clear C$_5$ symmetry resulting from a set of 1,019,938 particles and another map with slightly lower resolution and apparent C$_6$ symmetry from the remaining 398,806 particles. Both particle sets were subjected to one round of 2D classification and heterogeneous refinement before exporting them to Relion for per-particle CTF refinement and Bayesian polishing. Polished particles were imported back into cryoSPARC and cleaned up with a final round of 2D classification. The map corresponding to pentameric particles was then refined to a final resolution of 2.82 Å by homogeneous and non-uniform refinement with imposed C$_5$ symmetry. 3D variability analysis of the hexameric particle set revealed a particle cluster containing two helical densities in the central cavity, reducing the apparent C$_6$ symmetry to C$_2$. A heterogeneous

refinement with two classes enabled extraction of a particle set exhibiting these densities. This allowed subsequent homogeneous and non-uniform refinement of a final map to a resolution of 3.54 Å with imposed C$_2$ symmetry.

PO particles were selected by three rounds of 2D and two rounds of 3D classification with C$_5$ symmetry in Relion to discard non-protein contaminants and low quality classes. A final set of 947,286 particles with high-resolution features were further processed with per-particle CTF refinement and Bayesian polishing[39]. Polished particles were exported to cryoSPARC[29] to generate an initial 3D model that was refined to 2.97 Å by successive homogeneous and non-uniform refinement with imposed C$_5$ symmetry.

Final protein models were obtained by multiple iterations of manual model building in Coot 0.9.6[42], real-space refinement in PHENIX 1.19[43] and structure validation using MolProbity[44]. Maps and structural representations were prepared using Chimera v1.12[45], ChimeraX v1.3[46] or PyMol v2.3 (The PyMol Molecular Graphics System, Schrödinger).

## Blue native-polyacrylamide gel electrophoresis (BN-PAGE)

Purified GPR$_{\Delta 18}$ and GPR-A18L were buffer exchanged using a 0.5 mL 7 kDa molecular weight cut-off Zeba™ Spin Desalting Column (Thermo Scientific) equilibrated with Zeba buffer (20 mM HEPES-NaOH pH 7.5, 150 mM NaCl, 10% (v/v) glycerol, 0.4% (w/v) Cymal-5). Samples were then centrifuged at 20,000 x g and 4 °C for 5 min, and protein concentrations were determined spectrophotometrically using a Nano-Drop OneC (Thermo Scientific) at the absorption maximum of the retinal Schiff base (molar extinction coefficient of 45,000 M$^{-1}$·cm$^{-1}$).

For analytical BN-PAGE, ~5 µg of GPR$_{\Delta 18}$ or GPR-A18L were mixed with BN-PAGE sample buffer (50 mM Bis-Tris-HCl pH 7.2, 50 mM NaCl, 10% (v/v) glycerol), supplemented with 0.4% (w/v) Cymal-5 and 0.25% (w/v) Coomassie Brilliant Blue G-250 dye. Samples were loaded on a precast NativePAGE™ 4-16% Bis-Tris gel (Invitrogen) and run at 150 V for 60 min using an XCell SureLock™ Mini-Cell electrophoresis system (Invitrogen) filled with prechilled anode running buffer (50 mM Bis-Tris, 50 mM Tricine, pH 6.8) and dark blue cathode running buffer (anode running buffer containing 0.02% (w/v) Coomassie Brilliant Blue G-250). The latter was exchanged with light blue cathode running buffer (anode running buffer containing 0.002% (w/v) Coomassie Brilliant Blue G-250) and the run was continued at 250 V for 90 min. The gel tank was kept in ice water throughout the whole run. The gel was fixed by three rounds of briefly microwaving at 600 W and orbital shaking in fixing solution (40% (v/v) methanol, 10% (v/v) acetic acid) for 15-30 min. Finally, protein bands were stained using 25% (v/v) iso-propanol, 10% (v/v) acetic acid and 0.03% (w/v) Coomassie Brilliant Blue R-250 dye.

For preparative BN-PAGE and subsequent isolation of the GPR-A18L oligomers, 7.5 µg of purified GPR-A18L protein sample were mixed with supplemented BN-PAGE sample buffer (see above) and loaded into each of fifteen wells of a precast NativePAGE™ 4–16% Bis-Tris gel (Invitrogen). The run was performed as described above and the individual pentamer and hexamer bands were excised using a clean razor blade without prior fixation of the gel. Each set of fifteen excised bands per oligomer was submerged in 150–200 µL of Zeba buffer with 0.5% (w/v) Cymal-5 in a 2 mL Eppendorf tube and was incubated overnight at 4 °C. Extracted proteins were stored at 4 °C and a second round of gel extraction was performed, after which identical samples were pooled.

## UV-Vis spectroscopy and thermostability of isolated GPR-A18L oligomers

Gel-extracted GPR-A18L pentamer and hexamer samples were incubated with 17 µM all-*trans* retinal (from a 10 mM stock in ethanol) for 60 min at 4 °C to compensate for any potential loss of retinal during BN-PAGE. Samples were then transferred into a 0.5 mL 100 kDa

molecular weight cut-off Amicon® Ultra Centrifugal Filter Unit (Millipore) and subjected to six rounds of concentration (at 10,000 x g and 4 °C for 3 min) and dilution (to ~200 μL) with Zeba buffer to remove Coomassie Brilliant Blue G-250 dye. Six additional rounds of washing were performed in a new filter unit for optimised dye removal.

For spectral measurements, the washed and concentrated GPR-A18L pentamer and hexamer samples were diluted to 50 μL and transferred into a UV-Vis cuvette (UVette, Eppendorf). Twenty-four spectra were recorded for each oligomer using a NanoDrop OneC (Thermo Scientific), averaged and plotted in GraphPad Prism 10 software. For thermostability measurements, pentamer and hexamer samples were diluted to 7 and 3.5 μM, respectively, and thermostability was measured using a Prometheus NT.48 nanoDSF instrument (NanoTemper Technologies, Germany) in corresponding High Sensitivity capillaries. Thermal protein unfolding was performed between 40 and 95 °C at a heating rate of 1 °C per minute and the change in ratio of intrinsic protein fluorescence at 350/330 nm was tracked over time (excitation power = 100%). Data were plotted in GraphPad Prism 10 software and inflection temperatures of GPR-A18L pentamers and hexamers were computed by fitting the nonlinear regression function "Sigmoidal dose-response (variable slope)".

### Ligand extraction and mass spectrometry

5 μL of PO or GPR at 2 mg/mL (~75 μM) were added to 1 mL of a 200 mM hydroxylamine-HCl (Sigma Aldrich CAS: 5470-11-1, molecular weight: 69.49 g/mol) solution in 1:1 methanol/acetone. As a control, a 1 mg/mL decanoic acid (Sigma Aldrich W236403, >99.5%, molecular weight: 172.26 g/mol) solution was prepared in DMSO and, as for the protein samples, 5 μL were added to 1 mL of a 200 mM hydroxylamine-HCl solution in 1:1 methanol/acetone. All samples were incubated on an Eppendorf ThermoMixer Comfort 5355 and shaken for 4 h at 37 °C and 400 rpm to allow protein denaturation. Finally, 0.8 mL were used for a positive electron spray Q1 full scan (20–200 Da) on a hybrid triple quadrupole Sciex4000 QTRAP mass spectrometer (AB Sciex Concord, Ontario, Canada) equipped with an infusion pump (New Era syringe pump NE-300, S/N 300674) set to 15 μL/min infusion speed. The molecular ion peaks and fragment ions from Q2 were matched against the tandem mass spectrometry databases LipidBlast (fiehnlab.ucdavis.edu/projects/lipidblast) and Metlin (metlin.scripps.edu). The dot product score threshold was set as 0.7 and the minimum number of shared peaks (other than precursor ions) as 6 for high-confidence identification[47].

### LC-ESI-MS/MS methods

An IDA (Information Dependent Acquisition) method was used with two experiments, an EMS (enhanced MS) with a scan rate of 10,000 Da/s and a scan from 50 Da to 300 Da, and an EPI (enhanced product ion) with a scan from 50 Da to 200 Da, with precursor ion 188 $m/z$. A hybrid triple quadrupole Sciex5500 QTRAP mass spectrometer (AB Sciex USA) was used with Exion LC AC (AB Sciex) to perform the IDA with the following LC condition: The system was used in a positive-ion mode and a Turbo ion-spray with gas1, gas2, and curtain gas pressures set at 40, 70 and 20 psi, respectively. The source was heated at 650 °C. Sample temperature was maintained at 4 °C in the auto-sampler prior to analysis. For LC, a Reprospher HILIC-A, 5 μm, 150 × 4 mm (Dr. Maisch) column was used with a column guard, maintained at 40 °C. Mobile phases were: 5 mM ammonium formate in water with 0.1% (v/v) formic acid (solvent A) and 5 mM ammonium formate in MeOH with 0.1% (v/v) formic acid (solvent B). A flow rate of 0.5 mL/min in isocratic conditions was used with 30% solvent A and 70% solvent B. 10 μL of PO or GPR samples were injected into the machine to perform the LC and IDA acquisitions. Running time was set to 10 min.

### MRM method

To quantify N-hydroxydecanamide, we established the following multiple reaction-monitoring (MRM)-based LC-ESI-MS/MS method. Hybrid triple quadrupole Sciex4000 and Sciex5500 QTRAP mass spectrometers (AB Sciex Concord, Ontario, Canada) were used in conjunction with a Shimadzu UFLC (Shimadzu Corporation, Kyoto, Japan) and Exion LC AC (AB Sciex), repsecively, both featuring a cooled auto-sampler. The system was used in a positive-ion mode and a Turbo ion-spray with gas1, gas2 and curtain gas pressures set at 40, 70 and 20 psi, respectively. The source was heated at 650 °C. Quantification was performed in MRM mode and MRM conditions were optimized for all analytes and the internal standard palmitoylethanolamide-d4 (Cayman Chemical) by direct infusion. Sample temperature was maintained at 4 °C in the auto-sampler prior to analysis. For LC, a Reprospher HILIC-A, 5 μm, 150 × 4 mm column (Dr. Maisch, Switzerland) was used with a column guard maintained at 40 °C. The same mobile phases were used as above. A flow rate of 0.5 mL/min under isocratic conditions was used with 30% solvent A and 70% solvent B. Peaks were integrated and the Analyst software version 1.6.2 (AB Sciex Concord, Ontario, Canada) was used for quantification. Identification of compounds in samples was confirmed by comparison of precursor and product ion $m/z$ values and LC retention times with standards (N-hydroxydecanamide, >95% from Sigma Aldrich ENA408616050). The following MRM transitions were monitored for quantification of N-hydroxydecanamide: Qualifier (precursor ion 188 $m/z$ and product ion 104 $m/z$) and quantifier (precursor ion 188 $m/z$ and product ion 118 $m/z$) in positive polarity. Palmitoylethanolamide-d4 was used as internal standard with the precursor ion 305 $m/z$ and product ion 62 $m/z$ in positive polarity.

### Setup and analysis of MD simulations

Molecular dynamics simulations were performed to characterise the effect of different ligands on the dynamics of GPR (Fig. 4). The cryo-EM structures of GPR (PDB ID: 7B03) and PO (PDB ID: 8CNK) were used as initial models. Pentamers were embedded into a lipid bilayer composed of a 3:1 mixture of 1-palmitoyl-2-oleoyl phosphatidylethanolamine (POPE) and 1-palmitoyl-2-oleoyl phosphatidylglycerol (POPG), mimicking a bacterial membrane[48]. The systems were then solvated in a 20 Å water layer on both sides of the membrane, neutralised by the addition of 150 mM NaCl. The final systems measured 142 × 141 x 91 Å (Supplementary Fig. 18 and Supplementary Table 2). Initial assemblies were generated using the CHARMM-GUI webserver[49]. The force field parameters from a previous study on BPR[50] were used to build the retinal topology[51] with the psfgen plugin of VMD[52]. Protonation states of ionisable residues were assigned according to previous MD work on BPR and GPR[10,50,53–55]. Three systems starting from either the previous GPR or the current PO structure were assembled with (i) all-*trans* retinal bound to K232 via a protonated Schiff base, (ii) decanoate or (iii) no ligand in the chromophore binding pocket (see Table 1).

All simulations were run using the CHARMM36 force field[56], including CMAP corrections for the protein. Water molecules were described with the TIP3P water parameterisation[57]. Following the minimisation and equilibration protocols provided by CHARMM-GUI, the simulations were performed with OpenMM 7.7 molecular engine[58]. The non-bonded interaction cut-off was set to 12 Å with a switching distance at 10 Å. Periodic electrostatic interactions were computed using particle-mesh Ewald (PME) summation. Constant temperature of 300 K was imposed by Langevin dynamics with a damping coefficient of 1.0 ps and constant pressure of 1 atm was maintained with a Monte Carlo barostat[59]. The hydrogen mass repartitioning scheme was used to achieve a 4 fs time-step[60]. For each system, we performed one long simulation with a duration of 950 ns and two shorter simulations, each lasting 400 ns. We extracted and analyzed the final 500 ns of the longer simulation and the final 250 ns of the shorter simulations using

**Table 1 | Set-up parameters for MD simulations with GPR and PO**

| Starting structure | Oligomeric state | Ligand | Number of replica | Aggregated simulation time [μs] | Number of atoms |
|---|---|---|---|---|---|
| GPR[a] | Pentamer | All-*trans* retinal | 3 | 1.75 | 166,058 |
| GPR[a] | Pentamer | Decanoate | 3 | 1.75 | 165,973 |
| GPR[a] | Pentamer | None | 3 | 1.75 | 165,828 |
| PO[b] | Pentamer | All-*trans* retinal | 3 | 1.75 | 166,237 |
| PO[b] | Pentamer | Decanoate | 3 | 1.75 | 166,152 |
| PO[b] | Pentamer | None | 3 | 1.75 | 166,007 |

[a]PDB ID: 7B03.
[b]PDB ID: 8CNK.

VMD 1.9.3 and an in-house script implemented in tcl. The principal component analysis was computed using ProDy 2.2.0[61].

### Random acceleration MD (RAMD) simulations

Ten frames from each long simulation of decanoate and all-*trans* retinal were selected as initial conformations for the RAMD simulations. The simulations were conducted using the NAMD2[62] implementation with a default acceleration force of 16.0 kcal mol$^{-1}$ Å$^{-1}$[63]. Each simulation ran for a maximum duration of 2 ns. RAMD simulations were conducted separately for each ligand of each protomer, resulting in a total of 50 exit simulations per system.

### Reporting summary

Further information on research design is available in the Nature Portfolio Reporting Summary linked to this article.

## Data availability

Protein model coordinates and cryo-EM maps have been deposited in the Protein Data Bank (PDB) and Electron Microscopy Data Bank (EMDB) with the following accession codes: Pentameric GPR-A18L (8CQC, EMD-16795), hexameric GPR-A18L (8CQD, EMD-16796) and PO bound to decanoate (8CNK, EMD-16759). Cryo-EM raw data were deposited in the Electron Microscopy Public Image Archive (EMPIAR) under the deposition IDs: EMPIAR-12071 (GPR-A18L) and EMPIAR-12072 (PO). Source data are provided with this paper. Additional data related to the LC-MS experiments and molecular dynamics simulations are available at Zenodo.org (https://doi.org/10.5281/zenodo.12635820 and https://doi.org/10.5281/zenodo.12530050). Source data are provided with this paper.

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

## Acknowledgements

Pre-screening of cryo-EM samples was performed on equipment supported by the Microscopy Imaging Centre (MIC), University of Bern, Switzerland. For their excellent support to acquire cryo-EM datasets, we thank the Electron Microscopy Core Facility (EMCF) at the European Molecular Biology Laboratory (EMBL) and especially Felix Weiss. Molecular dynamics simulations were performed on UBELIX (http://www.id.unibe.ch/hpc), the HPC cluster at the University of Bern. This study was supported by the University of Bern, the UniBern Forschungsstiftung (grant 8/2023), the National Centre of Competence in Research (NCCR) Molecular Systems Engineering (grant no. 205608) and the Swiss National Science Foundation (SNSF; grants CRSII5_183481 and CRSK-3_190705). Stephan Hirschi is supported by a Postdoc. Mobility grant (no. 203042) by the SNSF.

## Author contributions

S.H., T.L., J.G. and D.F. conceived and designed the research. S.H., T.L., N.A., D.K., D.P. and Z.U. performed the experiments. S.H., T.L., D.P., J.G. and D.F. analysed the data. S.H., T.L. and D.F. wrote the manuscript. All authors contributed to manuscript revision and approved the final version.

## Competing interests

The authors declare no competing interests.
