## [Peer Review File · Nature Communications]

Structural insights into the mechanism and dynamics of proteorhodopsin biogenesis and retinal scavengingREVIEWER COMMENTS

Reviewer #1 (Remarks to the Author):

In this paper, Hirschi and colleagues present a cryo-electron microscopy structure of the green-light absorbing proteorhodopsin (GPR) in its pentameric, hexameric, and pentameric opsin forms. The achieved resolution using this method is undoubtedly a significant accomplishment. While there are multiple crystal structures of proteorhodopsins available in both pentameric and hexameric conformations, a longstanding debate revolves around which conformation represents the native functional state. Many biophysical studies, including EPR and AFM visualization, have shown evidence for the presence of both conformations simultaneously, or sometimes only one. Moreover, the authors themselves have reported several structures, potentially making this work appear less novel for publication in a journal like Nature Communications.

However, a notable strength of this paper is that the high-resolution pentameric and hexameric GPR structures obtained from a single construct strongly suggest the significance of the N-terminal signal peptide region in the hexameric arrangement. This result has the potential to provide answers to longstanding questions about why proteorhodopsins tend to form hexamers alongside pentamers.

Another remarkable strength of this study is the structural characterization of pentameric GPR opsin, which is generally challenging to obtain due to the inherent instability of opsins in the absence of retinal. The authors managed to get one with decanoate as a retinal substitute in their preparation. This information sheds light on how organisms lacking intrinsic retinal can maintain this protein. Additionally, through molecular dynamics (MD) simulations, the authors have provided insights into the potential entrance pathway of retinal.

While the structures presented in this paper exhibit reasonable quality and the study is well executed, I would like to address a few major critiques that could further enhance the manuscript:

1. Assuming that the pentamer/hexamer formation mechanism is accurate, it would be valuable to explore possible explanations for why proteorhodopsins tend to form higher oligomers (larger than hexamer) compared to other bacterial protein pumps. Are there any specific factors contributing to this behavior beyond what is known for other trimeric or pentameric proteins?

2. The influence of the hexameric organization on protein structure and retinal conformation is mentioned, but it would be beneficial to delve into how these structural alterations affect spectral features, such as the absorption maximum of each conformation. Is there any way to perform MD simulations specifically focusing on retinal conformation dynamics.

3. Investigate the extent of signal peptide cleavage in the native host environment to determine if hexamers are an artifact or possibly a temporary storage conformation, particularly when the protein is densely packed on the membrane.

4. Explore the potential effects of pH on multimerization, specifically for GPR with uncleaved signal peptide. Additionally, consider referencing relevant findings, such as the monomer-pentamer transition in the pentameric protein pump (<https://doi.org/10.1038/s41598-019-47445-5>).

5. Provide clarification on the source of decanoate and whether it could occur in the native environment without interfering with retinal binding. Address the feasibility and relevance of decanoate as a retinal substitute.

Reviewer #2 (Remarks to the Author):

Hirschi et al. use lipidomics to identify decanoate as a lipid bound to the model membrane proton pump proteorhodopsin when its native ligand retinal is absent (proteoopsin). They describe the structure of the protein-lipid complex via cryo-EM and molecular dynamics simulations. While I am much in favor of research looking to identify lipid ligands/cofactors of membrane proteins in general, as per my expertise, I will solely comment on the mass spectrometry part of this manuscript (primarily presented in Figure S6, p.13 and 29).

Overall and based on the evidence presented in the manuscript, I do not doubt that decanoate may serve as a ligand to proteoopsin. However, data to convince the reader whether decanoate is the major or even only such ligand is lacking. If the authors want to keep this strong statement, they should provide some evidence in this regard.

Identification of lipids from lipidomics experiments is commonly based on three pieces of information: 1. chromatographic retention time (ideally match with an authentic standard), 2. precursor m/z (ideally high resolution m/z allowing for determination of monoisotopic mass and thus molecular composition match), and 3. fragmentation spectra pattern (m/z and intensity ratios of characteristic peaks, ideally match with an authentic standard in both polarities).

The authors' comparison of the fragmentation pattern with a standard of decanoic acid is laudable, the intensities seem sufficient, and the assignment of possible fragments seems reasonable, even though a low resolution triple quadrupole mass spectrometer for tandem MS was employed and only operated in positive ion mode. It is unclear why no attempt of chromatography was performed (and the sample instead supplied via direct infusion), the retention time gained from which would help to confirm the identity of the lipid compared to the standard.

Most importantly, the authors do not provide information on the samples' major signals and their intensities overall. For a true ligand of the protein, we would expect i) a high absolute signal intensity in the PO sample and ii) a considerable intensity increase in the PO sample compared to the GPR sample.

The authors state that they only considered "signals found in PO but not in GPR" which is laudable since according to the hypothesis, the lipid should only be bound as ligand in the PO form. However, it is unclear which cut-off was used to determine whether a signal was found or not in the PO samples. This is important considering that trace amounts of the ligand may be present in either sample, which is why a ratio cut-off (i. e. 3-fold higher intensity in PO over GPR) may be better suited than the assumption of a lack of signal. How

many major signals (over a certain intensity threshold) were observed in total and what were their ratios between PO and GPR samples? Does this analysis support decanoate as the only or major potential ligand? Also, how many “multiple independent” experiments were performed that the identification is based on? (see page 13, line 7). As for the identification based on database search itself, was a dot-product score calculated? If so, what was it and what was the difference to the next match?

I further would encourage the authors to make the raw mass spectrometry data files available with their manuscript by depositing them to a repository.

Reviewer #3 (Remarks to the Author):

In this paper, Hirschi et al. use a combination of cryo-EM, mass spectrometry, and molecular dynamics simulations to provide new insights into green-light absorbing proton pump proteorhodopsin (GPR) assembly and retinal loading. First, they show that the N-terminal signal sequence dictates pentameric versus hexameric oligomeric assembly. Structures of an A18L mutant that prevents signal sequence cleavage show an approximately 3:1 ratio of hexamers to pentamers. The pentamer is essentially identical in structure to previously determined structures that lack N-terminal signal sequence. The hexamer, in contrast, has two signal sequence peptides threaded into the central cavity between subunits that expand the assembly and alter interactions of two protomers to accommodate the extra subunit. Second, they determine structures of retinal-free or proteoopsin GPR and find a “placeholder” molecule in the retinal binding pocket identified by mass spectrometry as decanoic acid. MD simulations show that decanoate binding is associated with increased retinal pocket access, that there are multiple feasible pathways for placeholder and retinal access/egress from the protein, and that other similar length acids and alcohols could serve as placeholders. Together, the study provides new and interesting insights into GPR structure, assembly, and retinal binding. The paper is very clear and well-written. The experimental structural work is performed and presented to a high standard. Issues that should be addressed prior to publication are listed below.

1. The functional consequence of hexameric versus pentameric assembly should be investigated. The structural characterization of different oligomeric states (and comparisons to previously determined structures) makes a convincing case for

unprocessed N-terminal signal sequences driving hexameric assembly through their incorporation into the central pore. The resulting structural differences are nicely explained and suggest the possibility of functional consequences. A comparison of wild type (signal sequence processed and pentameric) and A18L (unprocessed and predominantly hexameric) function would greatly strengthen the paper by providing insight into the question of whether oligomeric flexibility could have biologically interesting consequences.

2. There is at least one example of a retinal-free channelrhodopsin structure in the literature (ChRmine in Tucker et al., Nature Communications 2022). While no placeholder molecule was identified in the retinal binding site, it would be interesting to compare this to the proteopsin structure and MD results presented here. Are similar conformational changes in TMHs or retinal access pathways observed?

3. (Minor) Consider adding an additional arrow in the model figure indicating loss of the “placeholder” molecule for completeness and clarity.

4. (Minor) The resolution of structures presented here are all sufficient for the conclusions reached, but improved resolution may be achieved with further data processing (e.g., ML based particle picking, alternative approaches to classify 3D heterogeneity). The authors should consider depositing raw cryo-EM data to EMPIAR to facilitate potential reprocessing and/or software development efforts.

Reviewer #4 (Remarks to the Author):

The manuscript entitled "Structural insights into the mechanism and dynamics of proteorhodopsin biogenesis and retinal scavenging" submitted by Hirschi et al. reported revealing insights into the oligomeric assembly, variations in protomer stoichiometry and retinal incorporation, including a potential scavenging mechanism. To address these open questions of Microbial ion pumping rhodopsins (MRs), the authors executed a cryo-electron microscopy (cryo-EM) study and molecular dynamics (MD) simulations of the bacterial green-light absorbing proton pump proteorhodopsin (GPR) for model protein. The authors provided the first structural and dynamic insights into key steps of proteorhodopsin biogenesis, including i) elucidating the role of the signal peptide in the heterogeneous oligomerization of GPR, ii) uncovering the presence of decanoate as a temporary placeholder in retinal-free proteopsin, iii) exploring the potential of related molecules as placeholders, and iv) identifying possible pathways for the exchange of

decanoate and retinal, allowing the scavenging of the essential cofactor from the environment. This research is exciting and important for proteorhodopsin biogenesis and retinal scavenging. I can recommend being published in the "Nature Communications" with minor revisions. I also recommend that authors should re-check the manuscript carefully before submission. I hope the following suggestions and comments help the authors to improve the manuscript.

Comments:

1. The authors should add the total number of atoms and system sizes in each system for MD simulations. Cover shots of the system(s) are also suitable for understanding your MD systems easily.
2. I recommend the authors add an interaction diagram of retinal and ligand.
3. In conclusion, the authors proposed possible pathways for exchanging decanoate and retinal, allowing the scavenging of the essential cofactor from the environment (Fig. 6). This proposal is good enough. Still, the authors should state not in the conclusion section but in the discussion section. The conclusion should have consisted of the results of this research and enough discussion.
4. In Figure 4 (A-C), the authors show the principal component analysis of retinal, decanoate, and apo systems. The contributing rates of PC1s are the same as 21 %. Is it right? Contributing rates depend on the system, and the histogram does show it. Please check these. The authors should also offer each system's contributing rates of PC2.

Reviewer #5 (Remarks to the Author):

This manuscript reports on the structural elucidation of proteorhodopsin biogenesis, which has been clarify through the use of cryo-EM microscopy. Structural elucidation revealed the presence of electron density in the retinal binding site, which fail to correspond to retinal for several reasons (well sustained). Authors identified decanoic acid as a placeholder of retinal and protein stabilizer in the biogenetic process, and they also formulated an hypothesis of placeholder/retinal exchange by a set of MD simulations. Especially this latter element represent a novelty and opens to new speculations on the

biosynthetic pathway to generate proteorhodopsin or its homologues in the presence of a complex biological matrix.

Results have been generated by a robust and valid protocol, cryo-EM structures are of suitable resolution and further clarify the macromolecular assembly complexity and stoichiometry, although they fail to provide revolutionary novelty in the field. What is most important is the observation of a cofactor that is different from retinal in the binding site of proteorhodopsin. Indeed, the presence of this ligand triggered a number of speculations and hypothesis that have been partially addressed by MD simulations. Despite the rigor and the overall robustness of the computational protocol, the proof that a "potential mechanism by which the two molecules can be exchanged" is not fully convincing. Whether several exit pathways of decanoate and retinal have been identified, it is not clear if - in the authors' hypothesis - retinal recognizes the decanoate/proteorhodopsin complex or the apo proteorhodopsin (PO). Indeed, chemico-physical features of the proteorhodopsin binding site drastically change in the presence of decanoate, thus it will be important to provide additional (computational) details on the pathway and kinetics of exchange. Does the retinal displace decanoate? Or, does retinal bind to PO after decanoate left the protein through one of the possible exit pathways? This is an important point in the definition of the mechanism of molecules exchange that, in the opinion of this referee, constitutes the most appealing novelty of this work also in light of current drug design efforts against the homologue human rhodopsin.

To provide additional insights into the mechanism of exchange, authors might consider calculating some thermodynamics parameters from MD simulations, such as the delta energy of binding or the free energy of binding of non-covalent retinal (mimicking the recognition/approaching step) and decanoate. Moreover, to avoid confusion in the reader, it should be clearly stated that exit pathways of retinal have been simulated starting from its non-covalent adduct.

Methods are properly described and reproduction of authors' work is possible based on information provided. The only missing detail in the computational part is the protonation state of the Schiff base, as most of current works consider the protonated form of the Schiff base formed by interaction between the target K residue and retinal.

Reference citation should include also recent works in the field by different groups.

Point-by-point response for Hirschi et al. (NCOMMS-23-43494A)

Reviewer 1

In this paper, Hirschi and colleagues present a cryo-electron microscopy structure of the green-light absorbing proteorhodopsin (GPR) in its pentameric, hexameric, and pentameric opsin forms. The achieved resolution using this method is undoubtedly a significant accomplishment. While there are multiple crystal structures of proteorhodopsins available in both pentameric and hexameric conformations, a longstanding debate revolves around which conformation represents the native functional state. Many biophysical studies, including EPR and AFM visualization, have shown evidence for the presence of both conformations simultaneously, or sometimes only one. Moreover, the authors themselves have reported several structures, potentially making this work appear less novel for publication in a journal like Nature Communications.

However, a notable strength of this paper is that the high-resolution pentameric and hexameric GPR structures obtained from a single construct strongly suggest the significance of the N-terminal signal peptide region in the hexameric arrangement. This result has the potential to provide answers to longstanding questions about why proteorhodopsins tend to form hexamers alongside pentamers.

Another remarkable strength of this study is the structural characterization of pentameric GPR opsin, which is generally challenging to obtain due to the inherent instability of opsins in the absence of retinal. The authors managed to get one with decanoate as a retinal substitute in their preparation. This information sheds light on how organisms lacking intrinsic retinal can maintain this protein. Additionally, through molecular dynamics (MD) simulations, the authors have provided insights into the potential entrance pathway of retinal.

Authors: We thank the Reviewer for the detailed summary and appreciate the time invested in reading and reviewing our manuscript.

While the structures presented in this paper exhibit reasonable quality and the study is well executed, I would like to address a few major critiques that could further enhance the manuscript:

Authors: The points raised are addressed below. We are thankful for the feedback, which has contributed to the improvement of our manuscript.

1. Assuming that the pentamer/hexamer formation mechanism is accurate, it would be valuable to explore possible explanations for why proteorhodopsins tend to form higher oligomers (larger than hexamer) compared to other bacterial protein pumps. Are there any specific factors contributing to this behavior beyond what is known for other trimeric or pentameric proteins?

Authors: The literature reports various structural factors that govern the oligomeric state of microbial rhodopsins. These include the observed correlation of specific structural motives, including extended helices, 3-omega motif and flipped B-C loop, with the propensity to form particular oligomers (Morizumi *et al.* 2019, Sci. Rep.). Main cross-protomer interactions seem to be conserved among microbial rhodopsins forming trimeric arrangements (contributed by helices B and D) and pentamers (mediated by helices A, B and C). However, there are only minor differences regarding the oligomerisation interfaces, not involving the signal peptide, between green-light absorbing proton pump proteorhodopsin (GPR) pentamers and hexamers, and both oligomers share similar protomer structure.

We have added a brief discussion of the suggested categorisation of microbial rhodopsins based on these structural motives in the discussion section of the revised manuscript (see page 12) and referenced Morizumi *et al.* The structural data presented here agree with the proposed categorisation.

2. The influence of the hexameric organization on protein structure and retinal conformation is mentioned, but it would be beneficial to delve into how these structural alterations affect spectral features, such as the absorption maximum of each conformation. Is there any way to perform MD simulations specifically focusing on retinal conformation dynamics.

Authors: The authors agree that understanding the influence of oligomeric states on structure, stability and spectral properties is important and interesting. We believe that addressing this experimentally is the most effective approach. However, separating GPR pentamers from hexamers is very challenging, e.g., it is not feasible using size-exclusion chromatography (SEC). After an extensive investigation for possible separation methods, we successfully managed to separate hexamers and pentamers using blue native-polyacrylamide gel electrophoresis (BN-PAGE; Fig. S3A). This technique allowed us to extract the two oligomeric states from their respective gel bands, and subsequently measure their absorption spectra and thermostability. Our results indicate that the absorption maxima for the retinal chromophore in both hexamers and pentamers are not significantly different (Fig. S3B). Interestingly, however, the pentamers appear to exhibit higher stability compared to the hexamers (Fig. S3C). We believe that these novel experimental findings strengthen the manuscript and provide valuable insights into the relationship between oligomeric state, and protein function and structure.

3. Investigate the extent of signal peptide cleavage in the native host environment to determine if hexamers are an artifact or possibly a temporary storage conformation, particularly when the protein is densely packed on the membrane.

Authors: Unfortunately, the native γ -proteobacterial host of GPR has not been cultured yet and is therefore not available for this kind of experiment. As discussed in the conclusion, the signal peptide cleavage might either be relevant in the native host or an artefact of recombinant expression in *E. coli*. Additionally, we have included a short section at the end of page 8 in the revised version of our manuscript.

4. Explore the potential effects of pH on multimerization, specifically for GPR with uncleaved signal peptide. Additionally, consider referencing relevant findings, such as the monomer-pentamer transition in the pentameric protein pump (<https://doi.org/10.1038/s41598-019-47445-5>).

Authors: Gloeobacter rhodopsin (GR) was shown to exhibit a pH-dependent pentamer-monomer transition at low pH by Morizumi *et al.* The authors used SEC and electron microscopy to demonstrate that detergent-solubilised GR oligomers dissociate into monomers at pH 3. However, GPR expressing γ -proteobacteria are found in surface ocean waters (<https://pubmed.ncbi.nlm.nih.gov/11459054/>) with an average pH of 8 (<https://www.nature.com/articles/s41598-019-55039-4>). Therefore, it seems unlikely for GPR expressing γ -proteobacteria to encounter such acidic conditions in their natural environment. Given the natural pH conditions of GPR expressing γ -proteobacteria's environment, performing the suggested experiment at low pH appears to be outside the scope of our study, as it would not reflect the physiological conditions encountered by these organisms.

5. Provide clarification on the source of decanoate and whether it could occur in the native environment without interfering with retinal binding. Address the feasibility and relevance of decanoate as a retinal substitute.

Authors: In the native host, the most probable source for decanoate, or related molecules, are intermediates of bacterial fatty acid biosynthesis, including monocarboxylic acids with an even number of carbon atoms. Alternatively, decanoic acid or related molecules could be scavenged from the environment. We discuss now the most likely source for decanoate in the middle of page 25 of the revised manuscript.

Furthermore, we discovered that the presence of decanoate does not impede retinal binding or the formation of the Schiff base (Fig. S12). Considering this, decanoate emerges as a plausible substitute for retinal in the native host, with similar molecules offering viable alternatives (Fig. S11).

Reviewer 2

Hirschi et al. use lipidomics to identify decanoate as a lipid bound to the model membrane proton pump proteorhodopsin when its native ligand retinal is absent (proteopsin). They describe the structure of the protein-lipid complex via cryo-EM and molecular dynamics simulations. While I am much in favor of research looking to identify lipid ligands/cofactors of membrane proteins in general, as per my expertise, I will solely comment on the mass spectrometry part of this manuscript (primarily presented in Figure S6, p.13 and 29).

Overall and based on the evidence presented in the manuscript, I do not doubt that decanoate may serve as a ligand to proteopsin. However, data to convince the reader whether decanoate is the major or even only such ligand is lacking. If the authors want to keep this strong statement, they should provide some evidence in this regard.

Authors: The authors thank this Reviewer for the appreciation and the overall positive feedback. We agree that the Full Scan MS Q1 obtained with the extracted protein samples was of course not quantitative (see also reply below) and not the final proof that this lipid is the major ligand. Since we robustly saw the decanoate-derived *N*-hydroxydecanamide, which is the reaction product from decanoate and the hydroxylamine used to extract the lipids as the major molecule in the preparation in independent experiments, and based on this critique, we decided to generate an LC-MS method. Importantly, in the MS Q1 experiments *N*-hydroxydecanamide was not present in the GPR sample and retinal was not present in the PO sample but could be extracted and was seen in the GPR sample. To reply to this Reviewer's comments, providing tangible data, we have decided to develop and perform an LC separation and subsequent multiple reaction monitoring (MRM) scans, which in combination with a standard curve allowed for quantification and unambiguous determination of decanoate as major ligand (see below).

Identification of lipids from lipidomics experiments is commonly based on three pieces of information: 1. chromatographic retention time (ideally match with an authentic standard), 2. precursor *m/z* (ideally high resolution *m/z* allowing for determination of monoisotopic mass and thus molecular composition match), and 3. fragmentation spectra pattern (*m/z* and intensity ratios of characteristic peaks, ideally match with an authentic standard in both polarities). The authors' comparison of the fragmentation pattern with a standard of decanoic acid is laudable, the intensities seem sufficient, and the assignment of possible fragments seems reasonable, even though a low resolution triple quadrupole mass spectrometer for tandem MS was employed and only operated in positive ion mode. It is unclear why no attempt of chromatography was performed (and the sample instead supplied via direct infusion), the retention time gained from which would help to confirm the identity of the lipid compared to the standard.

Authors: We fully agree with this comment and have addressed it in the revised manuscript by reporting an additional LC-ESI-MS/MS-method using MRM. Data are described in the revised manuscript (2nd half of page 14) and presented in Fig. S8.

Most importantly, the authors do not provide information on the samples' major signals and their intensities overall. For a true ligand of the protein, we would expect i) a high absolute signal intensity in the PO sample and ii) a considerable intensity increase in the PO sample compared to the GPR sample.

The authors state that they only considered “signals found in PO but not in GPR” which is laudable since according to the hypothesis, the lipid should only be bound as ligand in the PO form. However, it is unclear which cut-off was used to determine whether a signal was found or not in the PO samples. This is important considering that trace amounts of the ligand may be present in either sample, which is why a ratio cut-off (i. e. 3-fold higher intensity in PO over GPR) may be better suited than the assumption of a lack of signal. How many major signals (over a certain intensity threshold) were observed in total and what were their ratios between PO and GPR samples? Does this analysis support decanoate as the only or major potential ligand? Also, how many “multiple independent” experiments were performed that the identification is based on? (see page 13, line 7). As for the identification based on database search itself, was a dot-product score calculated? If so, what was it and what was the difference to the next match?

Authors: The concern raised is valid since initially, we did not employ a quantitative approach or use an LC-MS method. Indeed, decanoate is present in high amounts (about equimolar to PO) and only in the purified PO, but not in the GPR (below the limit of detection), as confirmed by quantitative LC-MS. The experiments were performed in more than three independent samples and extractions: This is now stated on page 14. We set the dot product score threshold as 0.7 and the minimum number of shared peaks (other than precursor ions) as 6 according to (<https://www.nature.com/articles/s41467-017-01318-5>). This is now also stated in the manuscript (see *Methods* section, page 35).

I further would encourage the authors to make the raw mass spectrometry data files available with their manuscript by depositing them to a repository.

Authors: We have now made the raw data accessible at Zenodo.org (DOI: <https://doi.org/10.5281/zenodo.10995936>).

Reviewer 3

In this paper, Hirschi et al. use a combination of cryo-EM, mass spectrometry, and molecular dynamics simulations to provide new insights into green-light absorbing proton pump proteorhodopsin (GPR) assembly and retinal loading. First, they show that the N-terminal signal sequence dictates pentameric versus hexameric oligomeric assembly. Structures of an A18L mutant that prevents signal sequence cleavage show an approximately 3:1 ratio of hexamers to pentamers. The pentamer is essentially identical in structure to previously determined structures that lack N-terminal signal sequence. The hexamer, in contrast, has two signal sequence peptides threaded into the central cavity between subunits that expand the assembly and alter interactions of two protomers to accommodate the extra subunit. Second, they determine structures of retinal-free or proteopsin GPR and find a “placeholder” molecule in the retinal binding pocket identified by mass spectrometry as decanoic acid. MD simulations show that decanoate binding is associated with increased retinal pocket access, that there are multiple feasible pathways for placeholder and retinal access/egress from the protein, and that other similar length acids and alcohols could serve as placeholders. Together, the study provides new and interesting insights into GPR structure, assembly, and retinal binding. The paper is very clear and well-written. The experimental structural work is performed and presented to a high standard. Issues that should be addressed prior to publication are listed below.

Authors: We thank the Reviewer for the positive feedback. The mentioned aspects have been addressed below.

1. The functional consequence of hexameric versus pentameric assembly should be investigated. The structural characterization of different oligomeric states (and comparisons to previously

determined structures) makes a convincing case for unprocessed N-terminal signal sequences driving hexameric assembly through their incorporation into the central pore. The resulting structural differences are nicely explained and suggest the possibility of functional consequences. A comparison of wild type (signal sequence processed and pentameric) and A18L (unprocessed and predominantly hexameric) function would greatly strengthen the paper by providing insight into the question of whether oligomeric flexibility could have biologically interesting consequences.

Authors: We thank the Reviewer for this stimulating suggestion and agree that analysing the effect of the hexameric organisation on the spectral properties of GPR is very interesting. In order to address this question, we successfully separated pentameric and hexameric GPR by blue native-PAGE (BN-PAGE) and extracted them individually from the gel. This made it possible to perform spectrophotometric analysis of isolated pentamers and hexamers. The observed UV-Vis spectra show comparable absorption maxima for the retinal Schiff base in pentameric and hexameric GPR. Based on the similar spectral properties, this suggests no functional differences between the two oligomeric states.

In the revised version of our manuscript, we have incorporated these new results, i.e., BN-PAGE alongside spectra and thermostability profiles for pentamers and hexamers (Fig. S3) with discussions on pages 6, 11 and 12.

2. There is at least one example of a retinal-free channelrhodopsin structure in the literature (ChRmine in Tucker et al., Nature Communications 2022). While no placeholder molecule was identified in the retinal binding site, it would be interesting to compare this to the proteopsin structure and MD results presented here. Are similar conformational changes in TMHs or retinal access pathways observed?

Authors: We thank the Reviewer for bringing this to our attention. There seem to be only very minor differences between apo and retinal-bound ChRmine (RMSD 0.71 Å) in comparison to proteopsin (PO) and green-light absorbing proton pump proteorhodopsin (GPR) (RMSD 1.65 Å). This might indicate a higher rigidity, which might stem from mechanistic differences between channels and pumps, with the former needing less conformational flexibility. Alternatively, apo-ChRmine might be in a different conformation compared to decanoate-bound PO as no ligand is bound.

Interestingly, there is also an opening connecting to the retinal binding pocket in apo-ChRmine between transmembrane helices E and F, similar to PO and opsin. This supports our conclusion about the most likely retinal entry pathway found in our MD simulations and a general conservation of this mechanism.

We have discussed these results in the revised version of our manuscript in the corresponding paragraph (pages 20-22), referenced the findings of Tucker *et al.* and have updated Fig. S13 (previously Fig. S9) to include apo-ChRmine.

3. (Minor) Consider adding an additional arrow in the model figure indicating loss of the “placeholder” molecule for completeness and clarity.

Authors: An arrow indicating the loss of placeholder molecule was added to Fig. 6.

4. (Minor) The resolution of structures presented here are all sufficient for the conclusions reached, but improved resolution may be achieved with further data processing (e.g., ML based particle picking, alternative approaches to classify 3D heterogeneity). The authors should consider depositing raw cryo-EM data to EMPIAR to facilitate potential reprocessing and/or software development efforts.

Authors: We have made the raw cryo-EM data available on EMPIAR, see *Data availability*.

Reviewer 4

The manuscript entitled "Structural insights into the mechanism and dynamics of proteorhodopsin biogenesis and retinal scavenging" submitted by Hirschi et al. reported revealing insights into the oligomeric assembly, variations in protomer stoichiometry and retinal incorporation, including a potential scavenging mechanism. To address these open questions of Microbial ion pumping rhodopsins (MRs), the authors executed a cryo-electron microscopy (cryo-EM) study and molecular dynamics (MD) simulations of the bacterial green-light absorbing proton pump proteorhodopsin (GPR) for model protein. The authors provided the first structural and dynamic insights into key steps of proteorhodopsin biogenesis, including i) elucidating the role of the signal peptide in the heterogeneous oligomerization of GPR, ii) uncovering the presence of decanoate as a temporary placeholder in retinal-free proteopsin, iii) exploring the potential of related molecules as placeholders, and iv) identifying possible pathways for the exchange of decanoate and retinal, allowing the scavenging of the essential cofactor from the environment. This research is exciting and important for proteorhodopsin biogenesis and retinal scavenging. I can recommend being published in the "Nature Communications" with minor revisions. I also recommend that authors should re-check the manuscript carefully before submission. I hope the following suggestions and comments help the authors to improve the manuscript.

Authors: The authors thank the Reviewer for the positive feedback. The mentioned suggestions and comments have been addressed below.

1. The authors should add the total number of atoms and system sizes in each system for MD simulations. Cover shots of the system(s) are also suitable for understanding your MD systems easily.

Authors: We incorporated the additional information about the systems into Table 1 in the *Methods* section and included cover shots in the Supplementary Information (Fig. S15).

2. I recommend the authors add an interaction diagram of retinal and ligand.

Authors: An interaction diagram of the green-light absorbing proteorhodopsin (GPR) with retinal and proteopsin (PO) with decanoate has been added as a new Supplementary Figure (Fig. S9) and is now referred to in the corresponding *Results* section (page 16).

3. In conclusion, the authors proposed possible pathways for exchanging decanoate and retinal, allowing the scavenging of the essential cofactor from the environment (Fig. 6). This proposal is good enough. Still, the authors should state not in the conclusion section but in the discussion section. The conclusion should have consisted of the results of this research and enough discussion.

Authors: We see the Reviewer's point and have adapted the previous *Conclusion* text, i.e., we have split and adapted it into a *Discussion* and *Conclusion* part.

4. In Figure 4 (A-C), the authors show the principal component analysis of retinal, decanoate, and apo systems. The contributing rates of PC1s are the same as 21 %. Is it right? Contributing rates depend on the system, and the histogram does show it. Please check these. The authors should also offer each system's contributing rates of PC2.

Authors: Fig. 4 depicts the eigenspace defined by a principal component analysis (PCA) for the aggregated simulations of the retinal, decanoate and apo systems. For improved clarity, we have organized the results into separate panels for each system. The caption has been revised to explicitly state that the panels represent PCA of aggregated simulations. Furthermore, a dedicated PCA analysis for each simulation is now provided separately in the Supplementary information (Fig. S10).

Reviewer 5

This manuscript reports on the structural elucidation of proteorhodopsin biogenesis, which has been clarified through the use of cryo-EM microscopy. Structural elucidation revealed the presence of electron density in the retinal binding site, which fails to correspond to retinal for several reasons (well sustained). Authors identified decanoic acid as a placeholder of retinal and protein stabilizer in the biogenetic process, and they also formulated an hypothesis of placeholder/retinal exchange by a set of MD simulations. Especially this latter element represents a novelty and opens to new speculations on the biosynthetic pathway to generate proteorhodopsin or its homologues in the presence of a complex biological matrix.

Results have been generated by a robust and valid protocol, cryo-EM structures are of suitable resolution and further clarify the macromolecular assembly complexity and stoichiometry, although they fail to provide revolutionary novelty in the field. What is most important is the observation of a cofactor that is different from retinal in the binding site of proteorhodopsin. Indeed, the presence of this ligand triggered a number of speculations and hypotheses that have been partially addressed by MD simulations.

Authors: The authors thank the Reviewer for the feedback.

Despite the rigor and the overall robustness of the computational protocol, the proof that a "potential mechanism by which the two molecules can be exchanged" is not fully convincing. Whether several exit pathways of decanoate and retinal have been identified, it is not clear if - in the authors' hypothesis - retinal recognizes the decanoate/proteorhodopsin complex or the apo proteorhodopsin (PO). Indeed, chemico-physical features of the proteorhodopsin binding site drastically change in the presence of decanoate, thus it will be important to provide additional (computational) details on the pathway and kinetics of exchange. Does the retinal displace decanoate? Or, does retinal bind to PO after decanoate left the protein through one of the possible exit pathways? This is an important point in the definition of the mechanism of molecules exchange that, in the opinion of this referee, constitutes the most appealing novelty of this work also in light of current drug design efforts against the homologue human rhodopsin. To provide additional insights into the mechanism of exchange, authors might consider calculating some thermodynamics parameters from MD simulations, such as the delta energy of binding or the free energy of binding of non-covalent retinal (mimicking the recognition/approaching step) and decanoate.

Authors: We attempted to quantify the exchange and differences in binding affinity between decanoate and retinal through various computational approaches, including metadynamics, free energy perturbations and Gaussian accelerated molecular dynamics. Unfortunately, none of these strategies yielded convergent results.

In response to the Reviewer's insightful query about the mechanism of exchange, we acknowledge the importance of understanding whether retinal displaces decanoate or binds after decanoate exits. The experimental data supports the following conclusions: (i) the co-purification of decanoate with PO suggests a strong affinity, implying that retinal would need to displace decanoate. Additionally, (ii) the prolonged duration required for retinal incorporation (several minutes) hints at a comparable binding affinity between retinal and decanoate (Fig. S12).

Moreover, to avoid confusion in the reader, it should be clearly stated that exit pathways of retinal have been simulated starting from its non-covalent adduct.

Authors: We have clarified in the caption of Figure 5 that retinal is non-covalently linked for the exit pathways.

Methods are properly described and reproduction of authors' work is possible based on information provided. The only missing detail in the computational part is the protonation state of the Schiff base, as most of current works consider the protonated form of the Schiff base formed by interaction between the target K residue and retinal.

Authors: We simulated the protonated form of the Schiff base and added this important information to the *Methods* section (see *Setup and analysis of MD simulations*).

Reference citation should include also recent works in the field by different groups.

Authors: We have added the following references to our manuscript: <https://doi.org/10.1021/acs.jpcc.9b08189>, <https://doi.org/10.1002/chem.202200139> and <https://doi.org/10.1126/sciadv.adj0384>

REVIEWERS' COMMENTS

Reviewer #1 (Remarks to the Author):

The manuscript has been sufficiently revised according to reviewers' suggestions and comments properly by the authors. Now the manuscript is clear, concise, and well-written providing sufficient information for readers to follow the new findings of the present study. The methods are generally appropriate, and the biophysical and structural experimental results are clear and compelling with the generation of two possible multimeric state shifting. Also, their finding on retinal substitute decanoate as the temporary placeholder from the environment is remarkable and of significance to the field of retinal protein. Overall, the revisions are satisfactory, and I can recommend the manuscript be published in "Nature Communications".

Reviewer #2 (Remarks to the Author):

I laude the authors for the additional LC-MS/MS experiments performed. The chromatogram shown in Figure S8 for the PO sample looks well resolved and thus quantifiable and markedly lower in the GPR sample. For transparency, I would like the authors to add whether the chromatograms are shown for a single transition only? If so, which one (presumably quantifier $188.2 > 118.2$?), and what was the m/z range applied for the shown chromatograms?

Unfortunately, with the performed experiments, the authors did not fully address my concern regarding whether decanoate is the only or major potential ligand. While a semi-quantitative approach as the authors attempted, comparing ligand and protein quantities across two MS approaches, is adding value, my question was rather to show the comparison of full scan spectra between PO and GPR samples (and standard) as outlined in my initial review. By using an MRM method, the authors did not acquire this information with their LC method. The authors statement (p. 14) "Using LC-MS/MS, N-hydroxydecanamide was found only in the PO samples and could not be detected in the GPR Δ 18 samples (Fig. S8), thus confirming decanoate as the major ligand in the PO

samples.” is thus not fully accurate. The presented finding only confirms that the transition that presumably represents N-hydroxydecanamide was more intense in the PO compared to the GPR samples. It does not confirm this signal as the major signal found in the PO sample.

To understand better which peaks are unique to the PO sample, could the authors show a full scan spectrum for the GPR sample? Since this data was collected by infusion, the authors may also present an averaged spectrum for each sample. To improve interpretability, I may also suggest to draw the structures in Figure S7 as ions as they are observed in the MS. Due to the nature of decanoate as an acid, it is likely to show a stronger signal in negative ion mode, however, the authors chose positive ion spray and the N-hydroxydecanamide (188.2 m/z) species, although Figure S7 does not support calling the peak at 188.2 “major” as the authors state on p. 14. Why was only this peak chosen for the MRM method? Which structures do the authors propose for the fragment ions at 118.2 and 104.0 m/z?

As for quantitation, why was rolipram used as the only standard? While, as an internal standard, it serves as a rough extraction efficiency reference, the chemical structures and thus expected extraction and LC-MS behavior of rolipram and N-hydroxydecanamide are not particularly comparable. More importantly, the retention time of the two species (the suggested N-hydroxydecanamide and standard of rolipram) relative to each other is not very informative and does not add to the certainty of identification of the ligand in question. The sentence on p. 35 (“Identification of compounds in samples was confirmed by comparison of [...] LC retention times with standards”) is thus not correct - unless this data was not shown? It eludes the reviewer why the authors did not use the commercially available decanoic acid as an LC reference standard which they used in Fig. S7? We would expect identical retention times and transition efficiencies compared to the PO sample, allowing for added certainty in identification and quantitation. (As a note, even deuterated decanoic acid is available for purchase if the authors wanted an authentic internal standard.)

Lastly, I appreciate the authors uploading some data related to the LC-MS experiments to a public database, albeit - and in contrast to their statement in the response to reviewers - unfortunately not raw (.wiff) data. The file N-hydroxydecanamide_LC_;MS_Calibration and Results.xlsx that the quantification seems to be based on is a bit hard to digest without

additional explanation (e. g. why was once PO vs GPR compared and once Ret/no ret?). In addition, there seems to be no difference between the files Internal standards and Lev 11 preparation QTRAP4000.xlsx and LC-MS method N-hydroxydecamide_Internal standards and Lev 11 preparation QTRAP4000.xlsx?

Reviewer #3 (Remarks to the Author):

The authors have addressed all issues raised in my review in the revised manuscript. I have no further concerns.

Reviewer #5 (Remarks to the Author):

Authors addressed properly all my (and other reviewers') comments and doubts. The manuscript has notably improved accordingly. I have no further comments.

Point-by-point response for Hirschi et al. (NCOMMS-23-43494A)

Reviewer 2

I laude the authors for the additional LC-MS/MS experiments performed. The chromatogram shown in Figure S8 for the PO sample looks well resolved and thus quantifiable and markedly lower in the GPR sample. For transparency, I would like the authors to add whether the chromatograms are shown for a single transition only? If so, which one (presumably quantifier 188.2>118.2 ?), and what was the m/z range applied for the shown chromatograms?

Authors: We thank the Reviewer for the appreciation. We apologize that the methodology was not explained clearly enough and therefore provided a more detailed specification, also by adding more data. Supplementary Fig. 8 shows *N*-hydroxydecanamide only in the PO sample but not in the GPR sample. We used three transitions for the MRM method (188 m/z , 104 m/z and 118 m/z). While 188 m/z was used as qualifier ion, 104 m/z and 118 m/z were used as quantifier ions. To make it clearer that there is no ligand in the GPR sample, we have generated a figure more clearly illustrating the absence of a peak in the GPR sample by normalizing the intensity of the y-axis (Supplementary Fig. 8). It is important for us to clarify that there is no *N*-hydroxydecanamide in the GPR sample, which is a key finding of this experiment and strongly supports our conclusions (see also reply below). The MRM method does not use a m/z range (see also reply on the MRM method below).

Unfortunately, with the performed experiments, the authors did not fully address my concern regarding whether decanoate is the only or major potential ligand. While a semi-quantitative approach as the authors attempted, comparing ligand and protein quantities across two MS approaches, is adding value, my question was rather to show the comparison of full scan spectra between PO and GPR samples (and standard) as outlined in my initial review. By using an MRM method, the authors did not acquire this information with their LC method. The authors statement (p. 14) “Using LC-MS/MS, *N*-hydroxydecanamide was found only in the PO samples and could not be detected in the GPR Δ 18 samples (Fig. S8), thus confirming decanoate as the major ligand in the PO samples.” is thus not fully accurate. The presented finding only confirms that the transition that presumably represents *N*-hydroxydecanamide was more intense in the PO compared to the GPR samples. It does not confirm this signal as the major signal found in the PO sample.

Authors: In our initial MS scans, obtained via injection (to not lose any molecules in the LC), we identified decanoate (via its solvent reaction product *N*-hydroxydecanamide) as the most probable ligand that would fit the identified Coulomb density in the cryo-EM map (Figure 3C) and corroborate the MD simulations (Supplementary Fig. 12). Decanoic acid was only present in the PO sample but not in the GPR sample. Based on the comment of the Reviewer, we carried out an additional experiment, in which we employed the LC method used in the MRM to perform the requested scanning experiment on a Sciex5500 LC-MS QTRAP using the electron momentum spectroscopy (EMS) and enhanced product ion (EPI) mass methods. We added the scans and spectra to Supplementary Fig. 7 to illustrate that the molecule can be identified, as confirmed by the *N*-hydroxydecanamide standard. In agreement with the direct injection method, the PO and GPR samples differ in the presence of *N*-hydroxydecanamide as proven by the respective transitions in the EPI scan. The other peaks (not quantitative) do not match fragment ions corresponding to lipids that could fit the ligand volume of the PO pocket in the cryo-EM structure (Figure 3C). Therefore, we are confident that decanoic acid is a major ligand in our PO sample. As shown by the MD simulations (Supplementary Fig. 12), other small natural lipids could also act as placeholders for retinal, yet we did not identify those.

In addition: Regarding the extraction, we have tried different protein precipitation and extraction methods and validated them with retinal extraction, which we measured by LC-MS (not shown). We used hydroxylamine and the Folch extraction that we generally use for lipids as well as ethyl acetate plus hexane (9:1) with 0.1% (v/v) formic acid, which generally extracts a wide range of lipids. Only the hydroxylamine was able to extract retinal and decanoate (as solvent reaction products). Accordingly, we did not see a difference in the LC-based enhanced ion scan between the PO and GPR samples by using the Folch or ethyl acetate/hexane/formic acid methods (not shown). Only when we used hydroxylamine we could identify a difference, which was attributed to the extraction of decanoate, which reacts to *N*-hydroxydecanamide. Specific protein denaturation and solvent accessibility are the reasons for these findings. Hydroxylamine is known to destabilize microbial rhodopsins, as illustrated with the prototypical protein bacteriorhodopsin (Subramaniam et al. 1991, PNAS; Tokaji et al. 2010, Eur. Biophys. J.). This allows for the efficient extraction of the lipid retinal from rhodopsins, which tightly surrounds the cofactor. Consequently, the retinal, and in our case also decanoate, are efficiently released from the tightly packed protein. The fact that only hydroxylamine was able to extract the decanoate further substantiates that this lipid is in the pocket and not a surface lipid. A corresponding statement has been added to the revised manuscript (see page 12, bottom and page 13, top).

Since decanoic acid reacts with hydroxylamine to *N*-hydroxydecanamide, we focused on the latter molecule in the MRM method using the positive detection mode only. We hope that the Reviewer appreciates the high level of confidence coming from the MRM approach. Importantly, in the GPR sample no *N*-hydroxydecanamide was detected. Given the strength of the MRM methodology, we are persuaded that our statement is accurate.

To understand better which peaks are unique to the PO sample, could the authors show a full scan spectrum for the GPR sample? Since this data was collected by infusion, the authors may also present an averaged spectrum for each sample. To improve interpretability, I may also suggest to draw the structures in Figure S7 as ions as they are observed in the MS. Due to the nature of decanoate as an acid, it is likely to show a stronger signal in negative ion mode, however, the authors chose positive ion spray and the *N*-hydroxydecanamide (188.2 *m/z*) species, although Figure S7 does not support calling the peak at 188.2 “major” as the authors state on p. 14. Why was only this peak chosen for the MRM method? Which structures do the authors propose for the fragment ions at 118.2 and 104.0 *m/z*?

Authors: If we correctly understood the concern by the Reviewer, we now show an LC-based scan of the two samples, in addition to what we obtained by infusion. Also, we show the experiment, in which only the standard *N*-hydroxydecanamide is run in this method. These experiments are now shown in the revised Supplementary Fig. 7. The peak heights are not quantitative and only the presence of the peaks are important for the interpretation. The new figure supports the presence of 188 *m/z* in the EPI experiment. The fragment ions 118 *m/z* and 104 *m/z* are recombined ions and robustly generated by injecting the standard (*N*-hydroxydecanamide). This recombination can occur in the ion source or in the mass analyzer. There are different options for recombination that can be seen here: <https://mstools.epfl.ch/fragmentation/>, for instance C5H13NO (+) and C6H15NO (+) in Supplementary Fig. 7C.

As for quantitation, why was rolipram used as the only standard? While, as an internal standard, it serves as a rough extraction efficiency reference, the chemical structures and thus expected extraction and LC-MS behavior of rolipram and *N*-hydroxydecanamide are not particularly comparable. More importantly, the retention time of the two species (the suggested *N*-hydroxydecanamide and standard of rolipram) relative to each other is not very informative and does not add to the certainty of identification of the ligand in question. The sentence on p. 35 (“Identification of compounds in samples was confirmed by comparison of [...] LC retention times

with standards”) is thus not correct - unless this data was not shown? It eludes the reviewer why the authors did not use the commercially available decanoic acid as an LC reference standard which they used in Fig. S7? We would expect identical retention times and transition efficiencies compared to the PO sample, allowing for added certainty in identification and quantitation. (As a note, even deuterated decanoic acid is available for purchase if the authors wanted an authentic internal standard.)

Authors: We agree with this comment. Although rolipram is generally used as internal standard for a range of apolar to semipolar analytes, it may be suboptimal and we therefore used deuterated palmitoylethanolamide-d4, which is a more similar internal standard already available in our laboratory. This is now shown in the new Supplementary Fig. 8 using the more sensitive Sciex5500 QTRAP device. Importantly, the change of internal standard did not affect the quantification and again we obtained an approximate equimolar ratio of estimated PO to ligand. To increase the confidence of the quantification in addition to using *N*-hydroxydecanamide for the standard curve, we also used commercially available decanoic acid, which completely reacted with hydroxylamine to *N*-hydroxydecanamide for the standard curve. This led to the same quantitative results. The sentence on p. 35 is correct but the data was now shown before. This is now corrected by showing the *N*-hydroxydecanamide spike experiment (Supplementary Fig. 7C).

Lastly, I appreciate the authors uploading some data related to the LC-MS experiments to a public database, albeit - and in contrast to their statement in the response to reviewers - unfortunately not raw (.wiff) data. The file *N*-hydroxydecanamide_LC_;MS_Calibration and Results.xlsx that the quantification seems to be based on is a bit hard to digest without additional explanation (e. g. why was once PO vs GPR compared and once Ret/no ret?). In addition, there seems to be no difference between the files Internal standards and Lev 11 preparation QTRAP4000.xlsx and LC-MS method *N*-hydroxydecanamide_Internal standards and Lev 11 preparation QTRAP4000.xlsx?

Authors: We apologize for the confusion and the missing .wiff files. We have now uploaded corresponding files on zenodo.org: Please see *Data availability* statement for link. We also apologize for the confusion with nomenclature, as we internally referred to the PO sample as non-retinal and the GPR sample as retinal.